# The vacuole shapes the nucleus and the ribosomal DNA loop during mitotic delays

Emiliano Matos-Perdomo[1,2] (iD), Silvia Santana-Sosa[1,2] (iD), Jessel Ayra-Plasencia[1,2] (iD), Sara Medina-Suárez[1,2] (iD), Félix Machín[1,3,4] (iD)

**The ribosomal DNA (rDNA) array of *Saccharomyces cerevisiae* has served as a model to address chromosome organization. In cells arrested before anaphase (mid-M), the rDNA acquires a highly structured chromosomal organization referred to as the rDNA loop, whose length can double the cell diameter. Previous works established that complexes such as condensin and cohesin are essential to attain this structure. Here, we report that the rDNA loop adopts distinct presentations that arise as spatial adaptations to changes in the nuclear morphology triggered during mid-M arrests. Interestingly, the formation of the rDNA loop results in the appearance of a space under the loop (SUL) which is devoid of nuclear components yet colocalizes with the vacuole. We show that the rDNA-associated nuclear envelope (NE) often reshapes into a ladle to accommodate the vacuole in the SUL, with the nucleus becoming bilobed and doughnut-shaped. Finally, we demonstrate that the formation of the rDNA loop and the SUL require TORC1, membrane synthesis and functional vacuoles, yet is independent of nucleus–vacuole junctions and rDNA-NE tethering.**

## Introduction

One of the most remarkable visual events in cell biology is chromosome condensation, which takes place when cells transit from G2 into the prophase and metaphase (M phase). In higher eukaryotes, chromosomes are condensed over a central scaffold formed by topoisomerase II α, condensin I and condensin II (Paulson et al, 2021). Condensation requires the reorganization of the chromatin fiber into nested loops (Losada & Hirano, 2001; Gibcus et al, 2018; Paulson et al, 2021), so that chromosomes progressively shorten their length while becoming wider. The formation and expansion of these loops continues until chromosomes acquire their characteristic rod-like appearance by late metaphase.

The yeast *Saccharomyces cerevisiae* has served as an instrumental model for understanding fundamental processes of the eukaryotic cell. Its powerful genetic tools enable precise characterization of protein function. Condensin (there is a unique condensin complex in yeast) has not been an exception and has been scrutinized extensively (Strunnikov et al, 1995; Freeman et al, 2000; Lavoie et al, 2000, 2002; Bhalla et al, 2002). However, whereas in higher eukaryotes chromosome condensation is cytologically evident, this is not in yeast, where most of the nuclear mass remains visually amorphous throughout the cell cycle. The only exception to this is the ribosomal DNA (rDNA) array, located on the right arm of chromosome XII (cXIIr), and whose repetitive nature facilitates its visualization by FISH and fluorescence microscopy through specific rDNA binding proteins (e.g., Net1, Fob1, Cdc14, etc.) tagged with GFP variants. Previous studies showed that the rDNA of wild type cells appears unstructured in interphase, with a spotted and diffuse morphology by FISH, referred to as puff, and a crescent/oval shape at the nuclear periphery when labelled with GFP-tagged coating proteins (Guacci et al, 1994; Lavoie et al, 2002; Machín et al, 2005; Shen & Skibbens, 2017; Matos-Perdomo & Machín, 2018). As cells enter G2/M, this disorganized rDNA becomes a highly organized bar-like structure. In cells arrested in the late metaphase (sometimes referred to as mid-M in yeast) by the microtubule-depolymerizing drug nocodazole (Nz), the rDNA bar bends out of the rest of the chromosomal mass to become a horseshoe-like loop. This rDNA loop has been considered a condensed state of the repetitive locus and depends on active condensin for its establishment and maintenance (Freeman et al, 2000; Lavoie et al, 2002, 2004).

Condensin is not the only factor involved in the establishment and maintenance of the rDNA loop. Cohesin, which keeps sister chromatids together until anaphase, and the Polo-like kinase Cdc5 are also involved, yet their role is thought to regulate condensin activity on the rDNA (St-Pierre et al, 2009; Harris et al, 2014; Lamothe et al, 2020). Besides these, we previously reported that the rDNA loop requires an active Target of Rapamycin Complex 1 (TORC1) (Matos-Perdomo & Machín, 2018). TORC1 is the master complex that regulates cell growth and metabolism, controlling anabolic processes in the cell such as ribosome biogenesis, protein synthesis, and lipid synthesis (Wullschleger et al, 2006; Laplante &

[1]Unidad de Investigación, Hospital Universitario Nuestra Señora de Candelaria, Santa Cruz de Tenerife, Spain   [2]Escuela de Doctorado y Estudios de Postgrado, Universidad de La Laguna, Santa Cruz de Tenerife, Spain   [3]Instituto de Tecnologías Biomédicas, Universidad de La Laguna, Santa Cruz de Tenerife, Spain   [4]Facultad de Ciencias de la Salud, Universidad Fernando Pessoa Canarias, Santa María de Guía, Spain

Correspondence: fmachin@fciisc.es

Sabatini, 2009; Loewith & Hall, 2011). Previous works, including ours, have shown that TORC1 inactivation impinges on the morphology of the rDNA/nucleolus, reducing its size both in the interphase and mitosis (Tsang et al, 2003; Ha & Huh, 2011; Matos-Perdomo & Machín, 2018). Several mechanisms under the control of TORC1 have been proposed for this, including inhibition of both ribosome biogenesis and rDNA transcription, as well as autophagy of nucleolar components into the vacuole (Mochida et al, 2015; Mostofa et al, 2018; Matos-Perdomo & Machín, 2019). Because transcription of rDNA genes demands high levels of energy and resources, it is not surprising that the rDNA physiology is a main target of TORC1 (Li et al, 2006; Martin et al, 2006).

Yeasts undergo a closed mitosis and do not disassemble the nuclear envelope (NE) when entering M-phase (Thiry & Lafontaine, 2005). The rDNA is tethered to the NE throughout the cell cycle (Mekhail & Moazed, 2010; Taddei & Gasser, 2012). Importantly, previous works showed that both a prolonged mid-M arrest and mutants related to lipid metabolism lead to an enlargement of the NE that specifically affects the membrane region associated with the nucleolus and where the rDNA attaches to (Siniossoglou et al, 1998; Santos-Rosa et al, 2005; Campbell et al, 2006; Witkin et al, 2012). Here, we show that the vacuole, which occupies a large proportion of the cell volume, serves as a template to reconfigure the nuclear morphology during NE expansion in mid-M, and thus emerges as a major determinant in the morphology of the rDNA loop. The NE often acquires projections that contain the rDNA and distal parts of chromosome XII. These projections often bend themselves around the vacuole and reshape the nucleus towards a bilobed morphology, with one lobe containing most of the nuclear mass and with the nucleoplasmic handle that connects both lobes forming the rDNA loop. Alternatively, the rDNA in the projection opens up and blossoms into a horseshoe loop, leaving a NE ladle underneath. We further show that this reorganization of the nuclear shape requires an active TORC1 and new membrane synthesis but is independent of reported nuclear-vacuole and rDNA-NE contacts. We discuss how our new findings affect our vision of the rDNA loop as a model of chromosome condensation.

# Results

### The horseshoe loop is the most remarkable rDNA morphology in the mid-M arrest with nocodazole

By FISH, the mid-M (Nz-arrested) rDNA loop frequently appears as a horseshoe handle that bends out of the rest of chromosomes and makes the chromosomal mass to resemble a handbag (Guacci et al, 1994; Lavoie et al, 2002; Shen & Skibbens, 2017). The same pattern is often seen under the microscope when the rDNA is labelled with Net1-GFP and the nuclear mass with DAPI (Fig 1A) (Machín et al, 2005; Matos-Perdomo & Machín, 2018). Unlike FISH, fluorescence microscopy enables to see the loop in the context of an entire cell. Two features stand out above all and called our attention. First, the horseshoe loop goes across a significant proportion of the cell space; and second, it leaves a space under the loop (SUL) that is as large as the rest of the nuclear mass stained with DAPI, at the very

least. Thus, both the loop and the SUL often occupy a remarkable cell area, far beyond the proportional expectations for the nuclear/cytoplasmic volume ratios (Cantwell & Nurse, 2019; Deolal et al, 2021). At first, this suggests that either this volume ratio is dramatically shifted towards the nucleus or the nucleus is flattened to occupy a larger area and thus accommodate the enlarged rDNA loop. Hence, we aimed to study both the horseshoe loop and the SUL in detail.

It is important to note that the horseshoe loop is just one of the many morphologies of the rDNA array under the microscope (Fig 1B). Normally, these structures are seen after observation of flattened two-dimensional (2D) images (e.g., z-stack projections). However, three-dimensional (3D) reconstructions by confocal microscopy showed that, in many instances, non-horseshoe morphologies can correspond to actual horseshoe loops under different spatial perspectives (Fig 1C and further examples below). In addition, horseshoe loops are often bent in the shape of an arc when seen laterally (side views). Thus, the loop length measurements on 2D projections we present in this work should be always considered as under-approximations of real loop lengths.

In growing cells, horseshoe loops are rather infrequent, even in cells transiting through G2/M, where most rDNA arrays appear as small bars (Fig 1D and E). There are loops though; however, they are small (<2 µm in length) and appear packed (crossover and eight-like morphologies). After Nz treatment, and as cells begin to arrest in mid-M, loops surpass bars as the prominent morphology and the length of both is greatly increased (Fig 1D–F). In these circumstances, the bars appear to bend, and the packed loops open up to give rise to the horseshoe morphology, which is also more evident from different spatial perspectives as the horseshoe loop gets larger (Fig 1C–F). The remaining morphologies in budded S/G2/M cells, oval and resolved (two separated Net1 clusters in a single nucleus), seemingly correspond to cells that have not reached or have escaped from the mid-M arrest, respectively.

### Superresolution of the rDNA horseshoe loop shows that it is organized as a twisting thread of unequal density

We noticed that the rDNA loop does not appear homogeneously stained with different specific rDNA binding proteins such as Net1 and Cdc14 (Figs 1C and 2A). Z-stacking and deconvolution analysis showed segmented beads lengthwise the rDNA loop. We could discern up to nine of these beads or domains, with a median of five per loop (Fig 2B), which we could further confirm by Total Internal Reflection Fluorescence (TIRF) microscopy (Fig 2C). These beads likely represent the previously reported hierarchical organization of functional domains within the rDNA array (Wang et al, 2015; Dauban et al, 2019).

We next approached the fine characterization of the horseshoe loop by adding visualization through confocal superresolution microscopy (CSM). CSM showed that the rDNA loop was ~200 nm thick and comprised twisting of one or two threads, with sections resembling a spring (Fig 2D). We could also distinguish that the two-thread regions were separated by constrictions ("a" and "b" in Fig 2D). The temporal behavior of threads was rather dynamic, although the overall length and shape of the rDNA loop turned out to

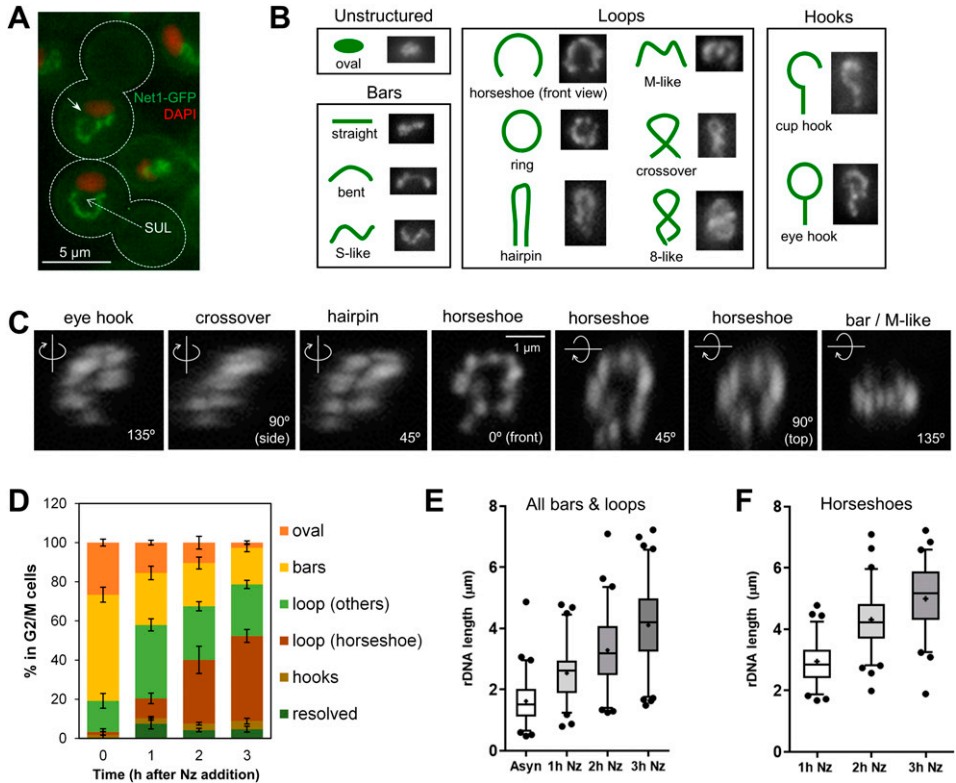

**Figure 1. Mid-M ribosomal DNA (rDNA) morphologies.**
**(A)** Prototypical examples of the rDNA loop in the mid-M (nocodazole) arrest. The rDNA is labelled by Net1-GFP and the main nuclear mass by DAPI. The dotted arrow points to the space under the loop (SUL). The white arrow on the upper cell points to a small gap observed sometimes on one flank of the loop. **(B)** Morphological patterns of the rDNA. Loops, especially the horseshoe loop, are common in mid-M blocks with Nz. **(C)** The morphology of the rDNA also depends on the visual perspective. A front view of a horseshoe loop (central image, 0°), captured by confocal superresolution microscopy and reconstructed in 3D from 0.15-μm-thick z-planes, was rotated on the y-axis (images to the left) or the x-axis (images to the right). **(D)** Proportion of rDNA morphologies during an Nz time-course experiment (mean ± SEM; n = 3). **(E)** Length of rDNA bars and loops during an Nz time-course experiment (n > 100 cells per condition). **(F)** As in (E) but only measuring horseshoe loops.

be quite stable over the course of short time-lapse video-microscopy (Video 1).

### The bases of the rDNA horseshoe loop are the flanking sequences of the rDNA array

We reasoned that the tracts of two twisting threads observed in some loops by CSM may comprise either the cohesed sister chromatids, resolved at 120 nm laterally, or domains of topologically packed/knotted rDNA units. In either case, the horseshoe loop would comprise the two sister chromatids travelling all along its length, with the first and the last unit of the repetitive locus at each edge of the loop (Fig 2E; "a" configuration). Alternatively, these two-thread tracts may expose the two halves of a coiled coil rDNA array (Fig 2B; "b"). In support that the latter ("b") could occur, we often found small gaps between one edge of the Net1 signal and the DAPI (Fig 1A, arrow). To differentiate between these two models, we combined into a single Net1-eCFP strain two bacterial *tetOs* arrays that we have previously designed to flank the rDNA array; *tetO:450* (rDNA proximal flank) and *tetO:487* (rDNA distal flank). This strain also carries the *tetO*-specific binding protein TetR-YFP. If the rDNA loop comprised an extended array that coils back ("b"), we should see both *tetOs* localizing at the same loop edge. On the contrary, if the rDNA follows the "a" configuration, we should see each *tetO* at each edge of the loop. We observed the "a" configuration in all cases (Fig 2F). This flanking positioning is in agreement with previous FISH data for the rDNA borders at the chromosome XII right arm (cXIIr) (Fuchs & Loidl, 2004). Aside from the frontal views of

horseshoe loops, the "a" configuration for the *tetO:450/487* partner was almost always seen for other rDNA loops and bars.

Similar configuration results were observed for horseshoe loops when we used a partner comprising *tetO:450* and a *tetO:1061*, which localizes near the cXIIr telomere. Even though the *tetO:1061* showed a looser localization, it was often close to the opposite *tetO:450* base (Fig 3A). The distance between the *tetO:1061* and the distal rDNA flank ranged from 0.2 to 4 μm (Fig S1A). In short time-lapse movies, the *tetO:450* remained immobile and attached to the loop base, whereas the *tetO:1061* moved rapidly and extensively (Video 2). Shorter distances to the nearest rDNA flank were measured for the *tetO:194*, which settles next to the chromosome XII centromere (Fig S1A). However, relative distances (μm/kbp) were the opposite, as the *tetO:194* is ~200 Kbs away from the proximal rDNA flank, whereas the *tetO:1061* is ~600 Kbs away from the distal rDNA flank (mean relative distances of 0.0025 and 0.0017 μm/kbp, respectively). The resulting mean apparent compaction ratio was 205 for the centromere-proximal flank tract, and 280 for the distal rDNA flank-telomere tract (Fig S1B). Previously, the axial compaction of chromosome arms in mid-M was estimated to be ~140 (Guacci et al, 1994). This implies that both tracts are probably curly or eventually fold back, especially the distal rDNA flank-telomere tract (Fig S1C; see below for further insights).

Next, we addressed the configuration of the rDNA horseshoe loop relative to the rest of nuclear DNA. To do so, we made use of the histone H2A2 (Hta2) labelled with mCherry to visualize the bulk of the chromatin. A close look at the Net1-GFP Hta2-mCherry strain showed that the Hta2 forms a protruding loop in ~55% of the

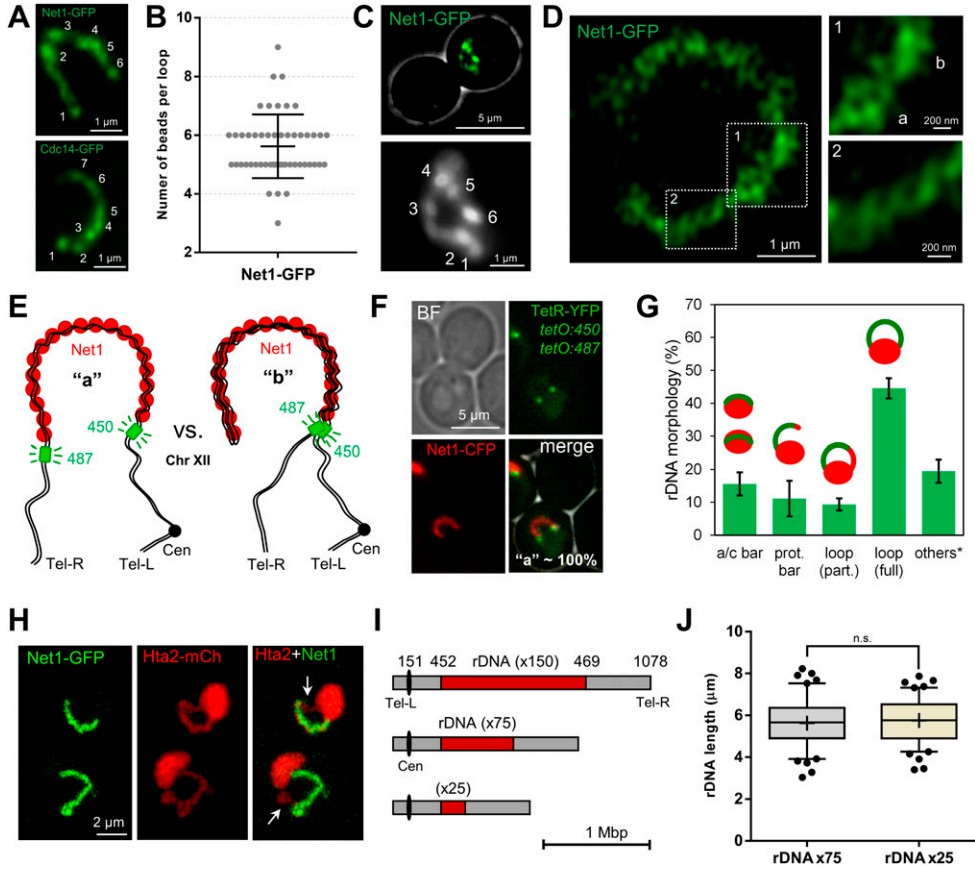

**Figure 2. Fine characterization of the mid-M ribosomal DNA (rDNA) horseshoe loop.**
**(A)** The rDNA loop comprises domains with distinctly enriched markers. Examples of rDNA loops with two specific markers: Net1 and Cdc14 (both deconvolved z-stack 2D projections). Numbers indicate counted "beads." **(B)** Counted beads for the Net1-GFP rDNA loop (n = 50 cells). Central horizontal solid line represents the mean and outer horizontal solid lines represent one SD. **(C)** The loop and its beads as seen by TIRF microscopy. **(D)** 120 nm superresolution of the rDNA loop. Inlet "1" features a section with two tight braided threads ("a" and "b") separated with a constriction. Inlet "2" features a spring-like section. **(E)** Schematics of two possible and mutually exclusive spatial configurations of the rDNA loop: (a), the loop has two bases, each comprising one of the rDNA flanks; and (b) the loop folds back, having both flanks residing on a single base. The green boxes indicate the *tetO* arrays inserted at 450 (rDNA proximal flank) and 487 (rDNA distal flank). **(F)** The rDNA loop in a strain with both 450 and 487 *tetOs*, the TetR-YFP and the Net1-ECFP (pseudo-coloured in red). All loops presented the configuration (a). BF, bright field. **(G)** Quantification of rDNA morphologies (Net1-GFP) relative to the main nuclear mass (Hta2-mCherry) as observed from z-stack 2D projections (mean ± SEM; n = 3); a/c, attached/crossing bar; prot bar, protruding bar. Others* mostly include oval and resolved into two rDNA signals. **(H)** Examples of partial horseshoe rDNA loops as seen by confocal superresolution microscopy (Net1 looks like a protruding bar but Hta2 forms a closed handle); white arrows point to where Net1 and the Hta2 do not overlap. **(I)** Scale drawing of chromosome XII with rDNA arrays bearing different numbers of its basic repetitive unit. The numbers above indicate Saccharomyces Genome Database (SGD) coordinates. **(J)** Length of rDNA loops from arrays with ~75 (n = 100 cells) and ~25 (n = 100 cells) copies.

Nz-blocked cells (Fig 2G). In about four fifth of these horseshoe loops, the rDNA (Net1-GFP) overlapped entirely with the Hta2 loop; however, in one fifth of the cases, the overlapping was partial, suggesting that flanking regions of the rDNA in cXIIr can belong to horseshoe loops (Fig 2H). Single-cell time lapse of these two presentations of the horseshoe loop showed they are interchangeable (Video 3; e.g., nuclei 2, 4, 6, and 13). These, together with the fact that attached/crossed bars (~15% of all Hta2/Net1 morphologies; Fig 2G) can represent top views of horseshoe loops (Fig 1C and Video 3, nuclei 3 and 15; Video 4, nuclei 3–5), imply that horseshoe loops are even more frequent than what can be observed at first glance in 2D projections. Another ~10% of Nz mid-M nuclei presented a protruding bar-like Hta2/Net1 signal, which appear to be anchored to the bulk of the nuclear mass through one base only, with no visible signs of they being perfect side views of horseshoe loops (yet a bunch of cases could be so; see Video 3, nuclei #5).

### The size of the rDNA array does not determine the length of the mid-M rDNA loop

Whereas it is undisputed that the rDNA is highly organized in mid-M, previous studies raised concerns about whether the loop is a condensed state of the locus, at least from a longitudinal point of view (Sullivan et al, 2004; Machín et al, 2005). We have repeatedly measured loop lengths in a Net1-GFP strain with ~150 copies of the 9.137-kbp rDNA unit and obtained mean values of ~5 μm (Fig 1F) (Matos-Perdomo & Machín, 2018). This translates into a relative length of ~275 kbp/μm (150 × 9.137/5). Considering that the length of one bp of naked B-form DNA is 0.34 nm, the estimated axial compaction ratio of the rDNA loop results in ~93 (275 × 0.34). In some instances, the loop reached ~9 μm (~152 kbp/μm; compaction ratio ~52) (Matos-Perdomo & Machín, 2018). Hence, the rDNA loop actually appears less compacted than a normal chromosome arm.

We questioned whether the loop length could be shortened by reducing the number of units of the rDNA array. We analyzed the rDNA length in two strains with the rDNA size fixed at around 75 and 25 copies (Figs 2I and S2A and B). This was achieved through the deletion of the *FOB1* gene, responsible for the change in size of the array (Kobayashi et al, 1998). Both strains were blocked in mid-M and the length of their rDNA loops was measured. We found no differences in their lengths (Figs 2J and S2C). The length was also equivalent to that of a ~150 copies rDNA (Fig 1E) (Matos-Perdomo & Machín, 2018). This implies that the compaction ratio of the rDNA can be lowered to, at least, ~10 (a loop of 25 copies reached 7.9 μm [0.34 × 25 × 9.137/7.9]). Theoretically, this packing ratio is close to

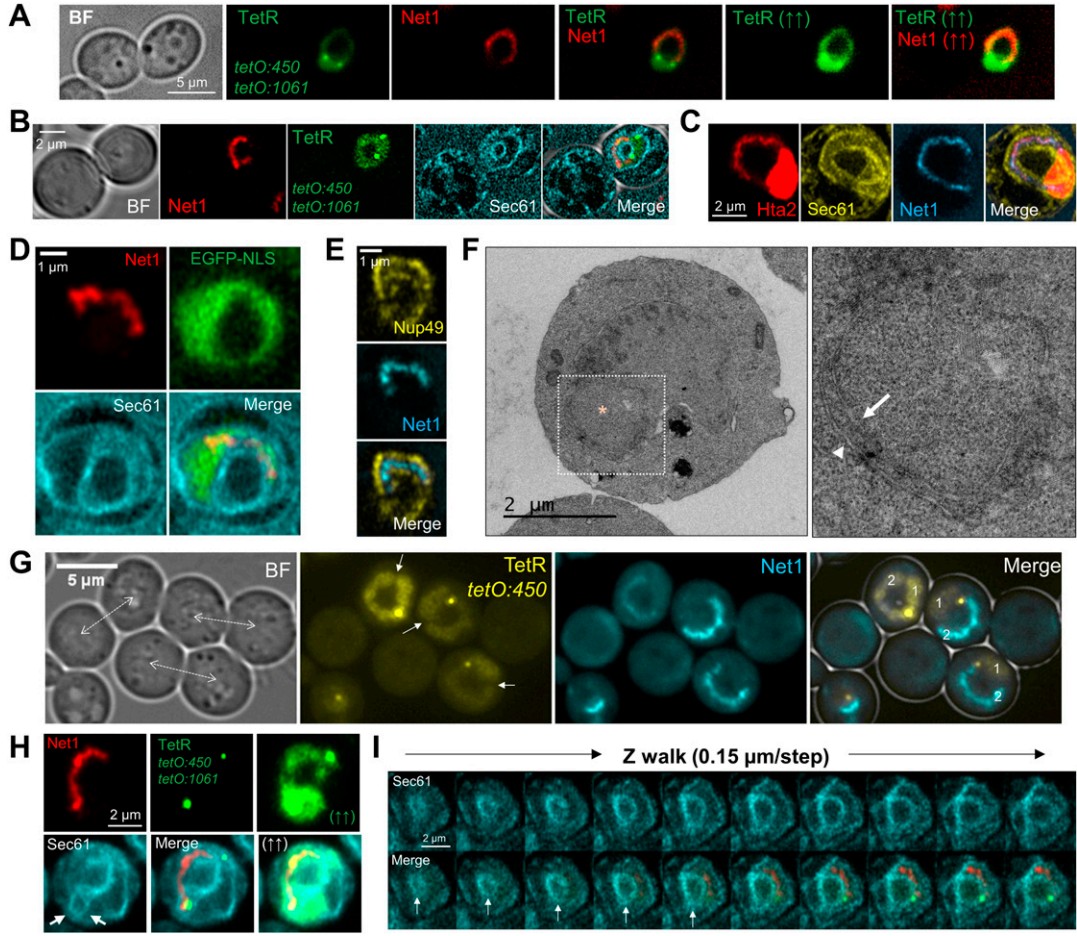

**Figure 3.  The space between the ribosomal DNA (rDNA) loop and the rest of the nuclear mass is not nuclear.**
**(A)** The heterologous TetR-YFP that freely circulates in the nucleoplasm does not label the SUL and can give the nucleus a doughnut-like appearance. The horseshoe loop comes from a strain with both 450 (proximal rDNA flank) and 1,061 (right arm telomere) *tetOs* (the two green dots), TetR-YFP and Net1-eCFP (pseudo-coloured in red). The pointing up arrows indicate saturated images. **(B, C, D, E, F)** The SUL is surrounded by nuclear membranes. **(B)** The nER (NE+ER) membrane marker Sec61 delimits both the internal and external boundaries of the nuclear space in the doughnut-like nucleus (TetR ring) that contains the rDNA loop (Net1-mCherry). **(C)** The space within the Hta2 handle that comprises the Net1 loop is surrounded by Sec61. Only the subset of confocal z-planes where the horseshoe loop was in focus were used for the z-projection (see Video 5 for the whole cell). **(D)** Like in (B) but with the nucleoplasm labelled with an EGFP-NLS construct instead of the TetR-YFP. **(E)** The rDNA loop (Net1-CFP) is also surrounded by the nuclear pore complex component Nup49. A single confocal z-plane is shown. **(F)** Transmission electron microscopy images of mid-M arrested cells with a nuclear morphology compatible with the ones shown in (B, C, D, E). The dotted square marks the area of interest, shown in more detail on the right. Electrodense material represents the nucleolus. The SUL is indicated by an asterisk. The arrow and the arrowhead point to the internal (handle-SUL) and the external (handle-cytosol) NE, respectively. **(G, H)** The SUL can be formed as a result of a highly bent bilobed nucleus. **(G)** Example of three mid-M cells (their polar axis indicated by a double dotted arrow) with bilobed nuclei, and where lobes touch each other at their poles (the TetR-YFP signal constrictions pointed by the white arrows), making the nucleus look like a doughnut. Each lobe is indicated by a number in the merged image. Note that the rDNA horseshoe loop forms the handle that connects both lobes and that one lobe contains the bulk of the nucleoplasm (1), whereas the rDNA extends through the other one (2). **(H)** A doughnut-like nucleus that results from both lobes overlapping at their ends (white arrows in the Sec61 image [see also Video 6, lowest cell]). **(I)** The SUL can contain a NE/ER ladle, which can be seen as a closing Sec61 ring while walking through the z-planes (white arrows) (see also Video 6, upper and mid cells). In micrographs: BF, bright field.

the 10 nm chromatin fiber, assuming that all the array was fully and periodically coated by nucleosomes (Felsenfeld & Groudine, 2003).

We conclude that (i) the size of the rDNA array does not determine the length of the rDNA loop, at least in a window of 25–150 copies; and (ii), conversely, the loop length is more or less fixed, with the compaction of the rDNA adapting to it, stretching the array if needed. Both the non-homogeneous compaction within the loop (beads) and its spring-like appearance described above may explain why the loop retains a constant length upon reducing the locus size.

## The space under the rDNA loop is not nuclear but shapes the entire nucleus

The second remarkable feature of the Nz mid-M arrest is the SUL, the large space between the loop and the main nuclear mass. To our benefit, in the strains that carry the TetR/*tetO* system not all TetR-YFP molecules bind to the *tetOs*. A pool of TetR-YFP freely circulates within the nucleus, labelling the nucleoplasm and, hence, marking the shape of the nucleus. When observing the Net1-eCFP horseshoe loops, the free TetR-YFP surrounded the rDNA in all cases. Strikingly, the SUL was void of TetR-YFP, especially evident in

frontal views of horseshoe loops (Fig 3A, see saturated TetR-YFP images, and Fig S3A). We hypothesized that this space may comprise other parts of the nucleolus, which could be somehow free of the TetR, perhaps through liquid–liquid phase separation (Feric et al, 2016; Matos-Perdomo & Machín, 2019). However, we found neither rRNA-processing proteins (Fig S3B) nor RNA (Fig S3C). In both cases, the mid-M arrest made these markers adopt a horseshoe-like morphology as well. Alternatively, the SUL could be filled up by part of the nucleus devoid of freely circulating proteins; for instance, heterochromatinized DNA. However, we did not find histone-coated (Hta2-mCherry) DNA, DAPI (after overexposing), or ultrasensitive YOYO-1 DNA stain within that space (Fig S3D and E).

In light of these negative labelling, we next questioned whether the SUL was nuclear. For addressing this, we added a third fluorescent marker to the previous set of strains, the ER membrane translocator Sec61, which labels the perinuclear ER, this being a continuum with the outer nuclear membrane of the nuclear envelope (NE). We found that the Net1 loop was surrounded both externally and internally by Sec61 (Figs 3B and C and S3F; see also Video 5 for a full appreciation of the spatial orientation of the second example). A similar pattern was obtained when an EGFP-NLS replaced TetR-YFP as the nucleoplasm reporter (Fig 3D). This was further confirmed by another NE marker, the nuclear pore complex (NPC) protein Nup49. Nup49 is more specific for the NE than Sec61, but its labelling shows a punctate distribution and is more challenging to interpret (Belgareh & Doye, 1997). Nonetheless, we could confirm that Nup49 labels the internal face of the loop as well (Figs 3E and S3G and H). To get further support for the claim that the SUL is not nuclear, we performed transmission electron microscopy (TEM). We found examples where the nucleus appears as a handbag, with a thin handle surrounded by a double membrane, and with the electrodense nucleolus close to or within this handle (Figs 3F and S3I).

The nucleoplasm labelling with both TetR-YFP and EGFP-NLS pinpointed that the nucleus is arranged as a ring-shaped doughnut, with a tendency to asymmetrically distribute the nucleoplasm towards a bulge, where most of the non-rDNA chromatin resides (Fig 3A–D). Frequently, a constriction in the bulge was evident (Fig S3J), which can be deep enough to separate the nucleoplasm signal in two (Fig 3G). 2D projections and 3D reconstructions of the NE (Sec61 and Nup49) showed that in these cases the bulge is made up of two lobes that overlap to various degrees (Figs 3H and S3J and K and Video 6). In addition, in most 3D reconstructions (80%; 16 of 20), the Sec61 signal distributed in depth as a hemisphere in the SUL (Fig 3I and Video 6, top two cells), suggesting that the NE (or the ER) could be ladle-shaped in lateral views, in addition to being doughnut-shaped in frontal views. This ladle was also seen with Nup49, which supports that it is made up of NE; however, Nup49 was much less abundant there, pointing out that NPCs may have restrictions to access the ladle (Fig S3H, arrows).

### The space under the rDNA loop is occupied by the vacuole

After realizing that the SUL was not nuclear, we added a cytosolic EGFP to a Sec61-eCFP Net1-mCherry strain to confirm the SUL was cytosolic. To our surprise, the EGFP signal was weaker in the SUL (Fig 4A). A key hint to understand the nature of the SUL came from the

visual comparison of the Net1 loop and the Hta2 handle with the entire mid-M cell as seen through the transmission light (bright field). We noticed that most large horseshoe loops appear to either entirely or partially surround the vacuole, which otherwise appears to sit on these loops in partial side views (Fig 4A–C). Hence, we checked whether the vacuole was occupying the SUL. We used several specific vacuolar markers, including the vacuolar lumen vital dyes Blue CMAC and carboxy-DCFDA, the vacuolar membrane (VM) non-vital dye MDY-64, and the VM marker Vph1-GFP. In all cases, at least one vacuole co-localized with the SUL (Figs 4D–I and S4A–D). There were instances where the co-localization of the SUL from a front view horseshoe loop and a sole vacuole was almost perfect, whereas in other cases the vacuole was too large for the SUL, or the SUL was filled with multiple smaller vacuoles (Fig S4A). VM markers such as Vph1-GFP gave the best signal at the equatorial central plane, which in large vacuoles was in a different z plane relative to frontal views of the rDNA horseshoe loop, making them to appear as if they were crossing the vacuole in 2D z-stack projections (Fig 4D). However, 3D reconstructions showed that part of the vacuole was sitting on the SUL (Fig 4G and H). With smaller vacuoles, at least one of them occupies the SUL (Figs 4E and I and S4A–D). Z-stack imaging, fluorescence intensity profiles and orthogonal projections of the vacuolar lumen confirm that the SUL was indeed occupied by these small vacuoles (Figs 4I and S4B–D).

### The rDNA horseshoe loop stems from small rDNA loops and bars that grow and bend around vacuoles

Next, we focused on the origin of the horseshoe rDNA loop with the vacuole in the SUL. To do so, we performed both time-course experiments and time-lapse video-microscopy after Nz addition. We observed that in an asynchronous population, most nuclei were spherical or slightly oval in cells transiting through S/G2/M (counting only the pre-anaphase budded cell subpopulation) (Figs 5A and S5A). Of note, in ~20% of these S/G2/M cells, the nucleus may appear squeezed between the vacuole and the plasma membrane, as if the NE is a malleable body that must seek allocation between two other stiffer bodies, the vacuole and the cell wall (Fig S5A, S/G2/M example). Shortly after Nz addition, budded cells elongate their nucleus (Fig 5A and Video 7), and this occurred with different degrees of symmetry in relation to the amount of nucleoplasm present along the extended nucleus (from pool noodles to finger-like projections; Fig S5B–G) and could carry or not primordial constrictions, which makes nuclei look like cashew nuts (Fig 5B, Video 7 and Video 8, and Fig S5B–F). These nuclear constrictions indicate that bilobulation could be a primordial event in the reshaping of the nucleus in Nz.

Nuclear elongation and constriction presentation could occur in two axes relative to the rDNA. The first and more abundant axis entails the spatial separation of the rDNA/nucleolus from the rest of the nuclear material, with the rDNA on one side or lobe (Fig 5B, upper two cells; Fig 5C and Video 7 and Video 8, upper cell). This configuration of having the nucleolus into protruding nuclear fingers/lobes was also confirmed by TEM (Fig S5B), and is in full agreement with the nucleolus-containing NE "flare" previously described by the Cohen-Fix's laboratory (Campbell et al, 2006; Witkin et al, 2012). In its early presentation, the rDNA often appears

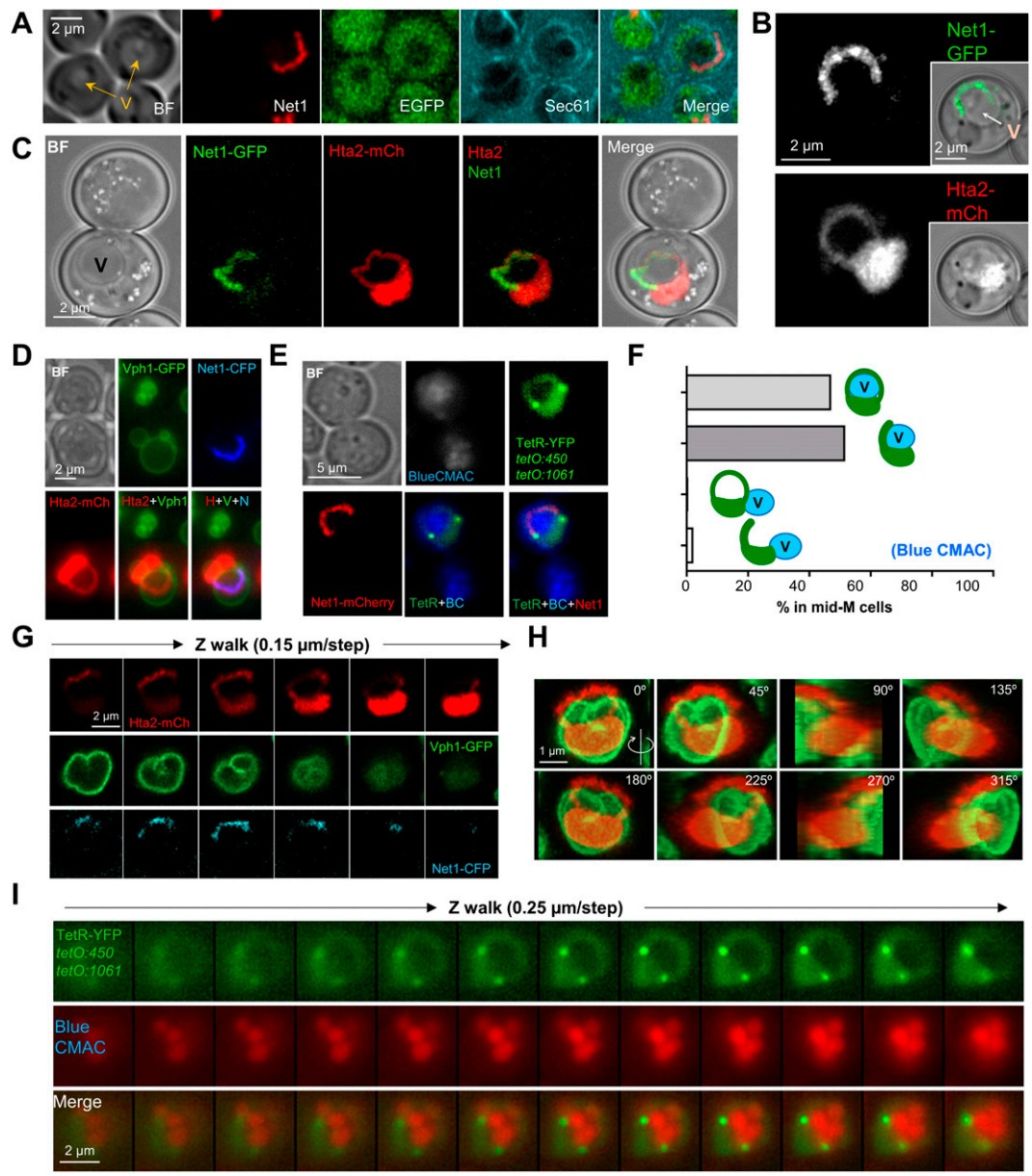

**Figure 4. The space under the ribosomal DNA (rDNA) loop is occupied by vacuoles.**
**(A)** Cytosolic EGFP weakly stains the SUL. The strain also bears Sec61-eCFP and Net1-mCherry to delimit the SUL (black holes within cell bodies). In the BF, vacuoles (V, pointed by arrows) may appear as balls of different density. Note how SULs and vacuoles colocalize. **(B, C, D, E, F, G, H, I)** Vacuoles reside in the SUL. **(B)** The rDNA loop (Net1-GFP and Hta2-mCherry handle) partly surrounds the vacuole (V), pointed with the arrow in the inlet BF image. **(C)** An example in which the vacuole (V in the BF) sits on a horseshoe loop. **(D)** A horseshoe loop, too small to engulf the vacuole (labelled with the vacuole membrane reporter Vph1-GFP), appears to cross the organelle in z-stack 2D projections. **(E)** Vacuole content (labelled with the vacuolar lumen vital dye Blue CMAC) is found in the SUL. **(F)** For a representative experiment as in (E), quantification of colocalization (for SUL) or juxtaposition (for nuclear flares) of rDNA bars and loops with vacuoles. Nuclear extensions ("flares") may correspond to either SUL side views or early stages before SUL formation (see below). **(G)** A walk through z planes of a horseshoe loop with a large vacuole on top. **(H)** A 3D reconstruction of (G) with serial 45° anticlockwise rotation on the y-axis. Note how the horseshoe loop leans onto the vacuole. **(I)** Z-plane walk-through of a TetR doughnut-like nucleus where vacuole lumens have been stained with Blue CMAC. In micrographs: BF, bright field; V, vacuole; BC, blue CMAC.

packed (oval or any of the small loop morphologies). However, as the nucleus becomes enlarged, this rDNA loop gets larger as well, either blooming into a horseshoe loop (compare the upper cell of Fig 5B with Fig 3C and Video 5; Video 9 for horseshoe bloom) or recoiling entirely one flank of cXII to become a protruding bar (see below for a detailed description of the latter).

The second axis of elongation leads to the formation of an rDNA bar that goes across the extended nucleus (Fig 5B and Video 8, lower cell). More evolved morphologies observed later in Nz suggest that these nuclei continue growing in length while bending, eventually forming a nucleoplasmic bridge that connects the two lobes (also as inferred from Video 7, lower cell). The rDNA is mostly

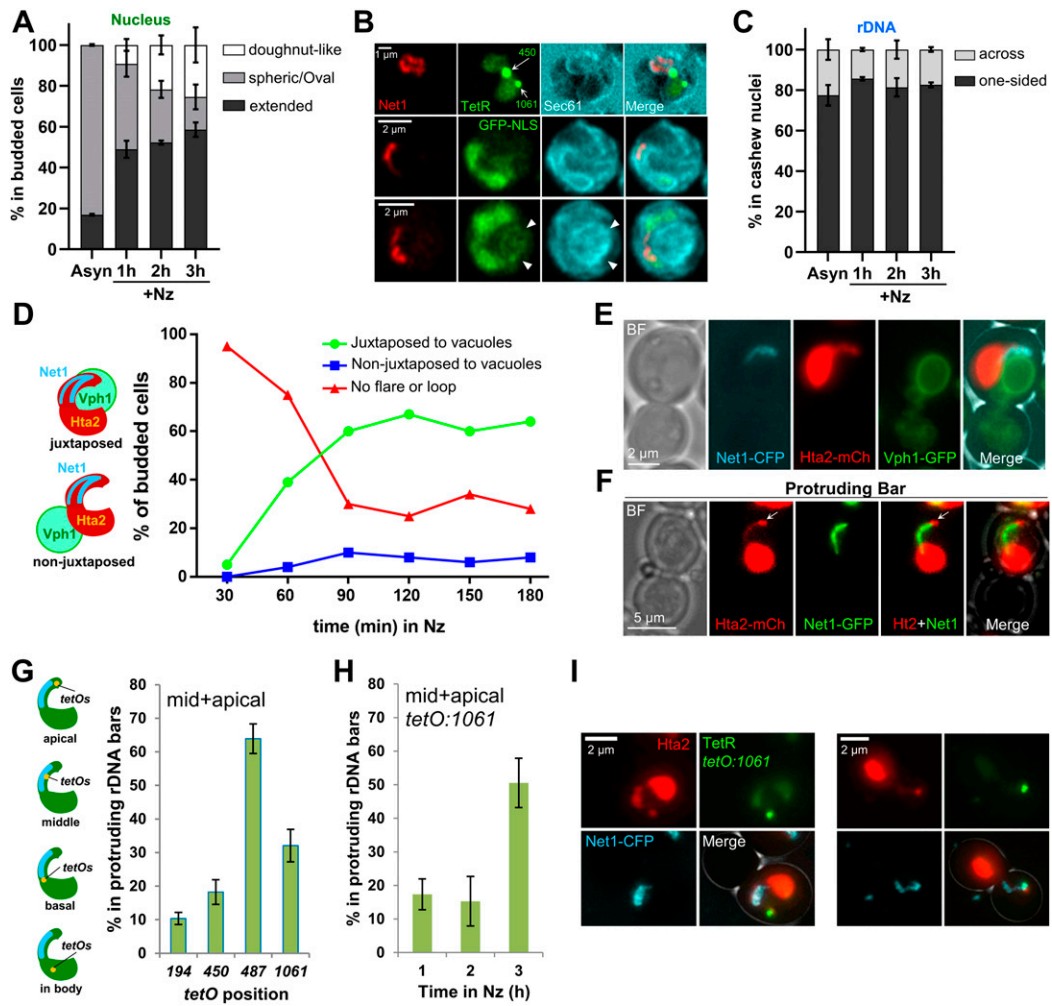

**Figure 5. The ribosomal DNA (rDNA) horseshoe loop stems from small bars and loops of rDNA that protrude out of the main nuclear mass and wrap around the vacuole.**
**(A)** Proportion of the three major nuclear morphologies observed during an Nz time-course experiment (mean ± SEM; n = 3); "extended" comprise a plethora of nuclear morphologies that deviate from the rounded nucleus observed in cycling cells. **(B)** Three examples of the most common extended morphology seen shortly after Nz addition, the cashew-like nucleus. In this morphology, a seemingly bilobed nucleus is seen, with lobes delimited by a primordial constriction. The rDNA can be located either entirely in one lobe, as a loop (upper cell) or a protruding bar (mid cell), or across both lobes (lower cell). **(C)** Proportion of the rDNA in one lobe (one-sided) or across both lobes in cashew-like nuclei during an Nz time-course experiment (mean ± SEM; n = 3). **(D)** Juxtaposition of growing nuclear extensions containing the rDNA and vacuoles during an Nz time-course experiment. "No flare or loop" (in red) indicates that neither extensions (finger-like or lobes) nor SUL were present. **(E)** An example of a protruding rDNA bar juxtaposed to a large vacuole. **(F)** An example of a protruding rDNA bar in the context of a protruding Hta2 bar. The arrows point to the denser Hta2 signal ahead of the Net1 tip. **(G)** Location of selected positions along cXIIr in protruding rDNA bars (from a pooled Nz time-course). "mid+apical" denotes that the corresponding *tetOs* array locates within the rDNA extension (mean ± SEM; n = 3). **(H)** Location of the cXIIr telomere in protruding bars during the Nz time-course (mean ± SEM; n = 3). **(I)** Two examples where the telomeric flank of the rDNA localizes ahead of the protruding rDNA bar, so that most of that chromosome arm (cXIIr) gets away from the nuclear mass. On the left, the protruding cXIIr remains in the mother; on the right, the protruding cXIIr crosses the neck. In micrographs: BF, bright field.

located in that bridge, acquiring the shape of a bent bar (Fig S5C). As the growth and bending continues, the two apical lobes can get closer until they touch each other and even overlap in 2D projections (Figs 3G and S5G and Video 6), resulting in a second class of horseshoe loops.

Strikingly, in cashew-like nuclei, which still lack a proper SUL, a NE ladle was often visible (Fig 5B, two lower cells; Video 8). Even traces of nucleoplasm (EGFP-NLS) were seen in this ladle (Fig 5B, arrowheads in the lowest cell), reasserting that the ladle must be formed by two close sheets of NE instead of ER. The early presence of this ladle points towards a germinal connection

between the NE subdomain associated to the rDNA and vacuoles. To determine when the interaction between the rDNA and the vacuole occurs, we again performed time-course experiments and time-lapse video-microscopy (Fig 5D and Video 10). We found that growing rDNA extensions tend to colocalize with vacuoles from the beginning, showing a constant ~7:1 ratio in favor of juxtaposition through the time course (Fig 5D and E). This germinal connection was also seen in time-lapse video-microscopy (Video 10).

A final hint to fully understand the nature and diverse origins of horseshoe loops came from observations that slightly differ from

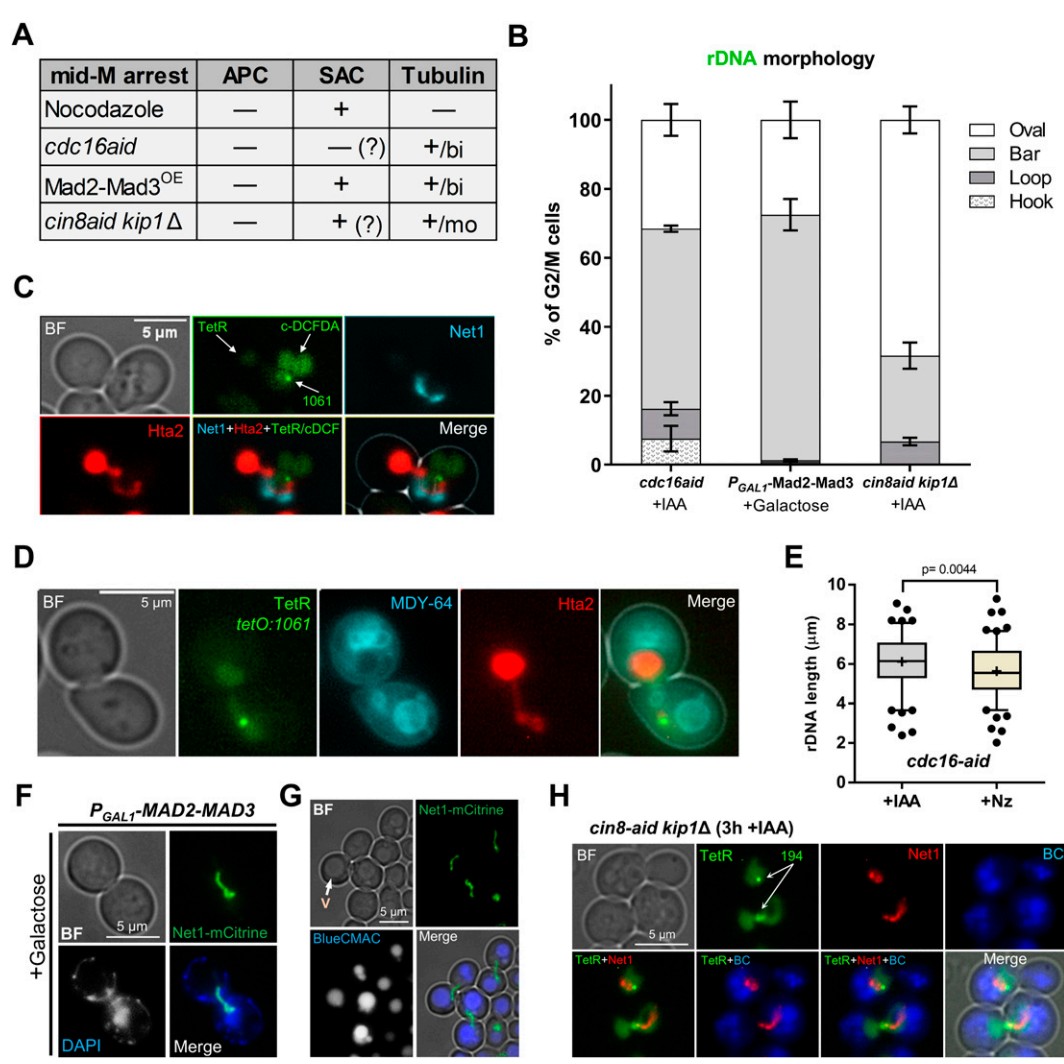

**Figure 6. The ribosomal DNA (rDNA) in mid-M arrests that preserve the microtubules.**
**(A)** Summary of the differences between four mid-M arrests (Nz, depletion of Cdc16, overexpression of the Mad2-Mad3 chimera, and depletion of kinesin motor proteins Cin8 and Kip1). APC, anaphase-promoting complex; SAC, spindle assembly checkpoint; OE, overexpression. The plus sign indicates either active (APC, SAC) or present (spindle); the minus sign indicates the opposite; the question mark indicates that the corresponding activation state is assumed, but there are contradictory data in the literature; bi, bipolar spindle; mo, monopolar spindle. **(B)** Morphology of the rDNA after the non-Nz mid-M arrests (mean ± SEM; n = 3). **(C, D)** Representative micrographs of the mid-M arrest observed after depleting Cdc16-aid. **(C)** An example with labels for the bulk of chromatin (Hta2), the rDNA (Net1), the nucleoplasm (free TetR-YFP), the cXIIr telomere (tetO:1061) and the vacuolar lumen (c-DCFDA), In the green channel, the three latter structures are labelled and differentiated by intensity and morphology; nucleoplasm is the weak signal on the left, the cXIIr telomere is the spot, and the vacuolar lumen is the stronger signal on the right. **(D)** Like in (C) but with the vacuolar membrane labelled with MDY-64. The strain is the same, but Net1-CFP is completely masked by the much stronger MDY-64 signal. Note in both examples the nuclear disposition across the neck, with the rDNA and the bulk of the DNA mass residing in different cell bodies, and how the rDNA interacts with vacuoles in the receiving body. **(E)** Length of the rDNA (Net1) in Cdc16-aid plus IAA (>100 cells) versus Cdc16-aid plus Nz (>100 cells). **(F, G)** Representative micrographs of the neck-crossing rDNA bars observed after overexpressing Mad2-Mad3. **(F)** Net1 together with DAPI staining. **(G)** Net1 together with BC staining. **(H)** Representative micrograph of the mid-M arrest observed after depleting Cin8-aid in a kip1Δ background. The upper cell shows a case where the nucleus and the rDNA remain in the same cell body; note that there is no horseshoe loop. The lower cell shows a neck-crossing rDNA bar, as in other non-Nz mid-M arrests. In micrographs: BF, bright field; V, vacuole; BC, Blue CMAC.

the morphological patterns described above. As stated, the horseshoe rDNA loop is the most remarkable morphology of the rDNA array in mid-M, but protruding rDNA bars are observed as well in 10% of the arrested cells (Figs 2G and 5B, mid cell). These protruding bars often contained a brighter Hta2 spot at the tip (Fig 5F, arrow), which must correspond to one of the flanks of cXII, probably up to the corresponding telomere. To precisely determine the arrangement of the chromosome in these protruding bars, we looked at the location of the four tetOs along cXIIr (Fig 5G). We

observed that the distal flank of the rDNA (tetO:487) tended to be significantly present in the nuclear projections (~65% of the cases, mostly in an apical position), although in ~18% of the bars we found the proximal flank (tetO:450) in there; and even the centromere in ~10% of bars. The fact that the distal rDNA flank was more frequently found than the proximal flank suggests that distal cXIIr regions are more prone to get into the growing extension. Compared with the rDNA distal flank, we found fewer examples of the cXIIr telomere (tetO:1061) in the projection, strongly pointing out that about half of

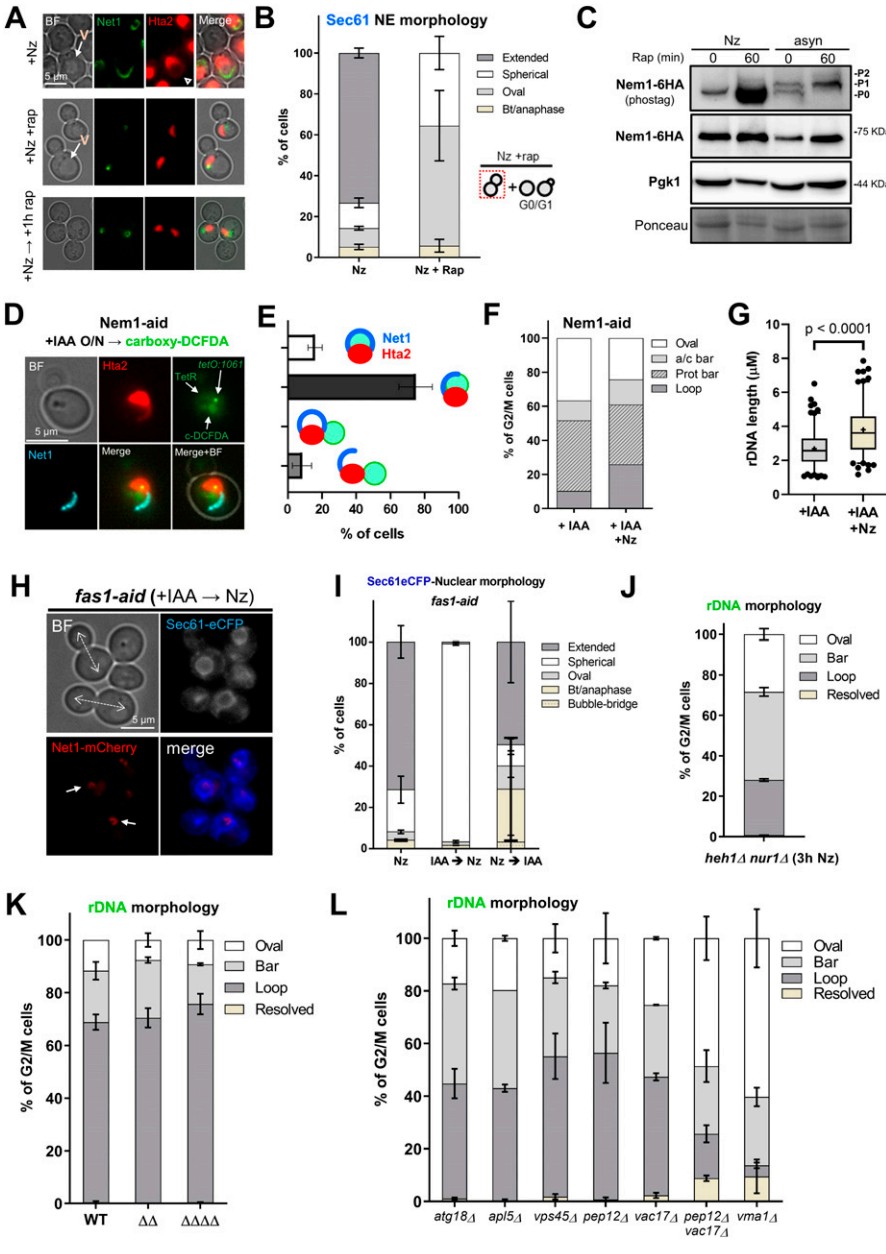

**Figure 7. The ribosomal DNA (rDNA) loop requires TORC1, membrane synthesis and functional vacuoles.**

**(A, B)** Rapamycin prevents formation and maintenance of the rDNA loop in Nz-treated cells.

**(A)** Representative cells upon Nz alone, Nz plus rapamycin co-treatment, and 1 h rapamycin after a previous Nz mid-M arrest. The arrowhead points to the Hta2 handle in the Net1 horseshoe loop in Nz alone. **(B)** Morphology of the NE (Sec61) in Nz and Nz plus rapamycin (mean ± SEM; n = 3; only mid-M cells counted). **(C, D, E, F, G)** Over-synthesis of NE leads to nuclear and rDNA extensions morphologically distinct to those observed in Nz mid-M arrest. **(C)** Western blot of Nem1-6HA under Nz alone and Nz followed by 1 h rapamycin. An equivalent rapamycin regime over an asynchronous culture was performed as well for reference. The position of the three expected bands is indicated in the electrophoresis run with Phos-tag (top image). The others were standard SDS–PAGE controls. **(D)** A representative G2/M cell in asynchronous cultures growing with a depleted Nem1. Note it comprises a seeming protruding bar, which is actually a closed cXIIr loop as deduced from the cXIIr telomere position, into a finger-like flare nuclear projection. Five subcellular structures are labelled: the bulk of chromatin (Hta2), the rDNA (Net1), the nucleoplasm (free TetR-YFP), the cXIIr telomere (*tetO: 1061*), and the vacuolar lumen (c-DCFDA), In the green channel, the three latter structures are labelled and differentiated by intensity and morphology; nucleoplasm is the weak signal on the top that colocalizes with Hta2, the cXIIr telomere is the spot, and the vacuolar lumen is the stronger signal underneath. **(E)** Quantification of rDNA morphology (loop versus bars) and vacuole juxtaposition in asynchronous cells with depleted Nem1 (mean ± SEM; n = 3). **(F)** Quantification of rDNA morphologies in a representative experiment where Nz was added to cells depleted of Nem1. Note that the proportion of loops doubled after Nz, yet it does not reach that of Nz alone (from 40% to 70% in other Nz experiments in this work). **(G)** Length of the rDNA loops and bars in cells depleted of Nem1 before and after Nz addition. Note the additive Nz effect. **(H, I)** Depletion of the fatty acid synthetase beta subunit Fas1 prevents nuclear extension upon Nz treatment. **(H)** Representative micrograph with two mid-M cells (whose polar axes are indicated by double dotted arrows in the BF) in which the NE is round and the rDNA are small bent bars (white arrows). **(I)** Morphology of the NE (Sec61) in Nz (control), IAA → Nz (establishment) and Nz → IAA (maintenance) (mean ± SEM; n = 2; only mid-M cells counted). **(J)** Quantification of the rDNA morphology in the Nz mid-M arrest in a mutant that does not tether the rDNA to the NE (mean ± SEM; n = 2). **(K)** Quantification of rDNA morphologies upon Nz in mutants for nucleus-vacuole junctions (mean ± SEM; n = 2). Nucleus-vacuole junction mutants: ΔΔ, *mdm1Δ nvj3Δ* double mutant; ΔΔΔΔ, *mdm1Δ, nvj1Δ nvj2Δ nvj3Δ* quadruple mutant. **(L)** Quantification of the rDNA morphology in the Nz mid-M arrest in mutants that affect vacuole size, inheritance, and function (mean ± SEM; n = 2). In micrographs: BF, bright field; V, vacuole.

the rDNA protruding bars are still cXIIr loops (likely side views of the partial rDNA loops we reported in Fig 2G and H). However, as Nz incubation goes by, the cXIIr telomere did move to the tip (Fig 5H and I). Remarkably, these rDNA bars, and their surrounding nucleoplasmic flares, showed extensive bending, which were clearly reminiscent of incomplete states of the horseshoe loop and the overlapping bilobed nucleus described above (Fig 5I. left nucleus; Fig S5C and G); there were also cases where the protruding bar crossed the neck into the bud (Fig 5I, right nucleus).

## The horseshoe rDNA loop requires the absence of microtubules

Previous works demonstrated that nuclear extensions and flares are characteristic of cells blocked in mid-M (Witkin et al, 2012). Nz is the most common experimental tool to achieve the mid-M arrest. Nz depolymerizes microtubules, which dismantles the spindle apparatus and activates the spindle assembly checkpoint (SAC) (Jacobs et al, 1988; Hoyt et al, 1991). With an active SAC, the anaphase-promoting complex (APC) activator Cdc20 is tightly bound and inhibited by the SAC components Mad2 and Mad3

(Peters, 2006; Musacchio & Salmon, 2007). Considering the effect of Nz on microtubules and the SAC, we next chose to study the effect of arresting cells in mid-M by other means. We planned two different strategies that preserved the bipolar spindle, yet they differ in the activation state of the SAC (Fig 6A). On the one hand, we depleted Cdc16, an essential component of the APC, by creating a *cdc16-aid* conditional allele. We used this strategy because thermosensitive alleles require incubation at 37°C, and we and others previously showed that this triggers a mild heat stress response that impinges on the rDNA loop (Shen & Skibbens, 2017; Matos-Perdomo & Machín, 2018). Degradation of Cdc16-aid can be triggered by adding the auxin indole-3 acetic acid (IAA) to the medium (Fig S6A and B) (Nishimura et al, 2009). Under this condition, the APC can be inactivated without interfering with the spindle and/or the SAC, although there is a report about a possible activation of the SAC upon APC inactivation (Lai et al, 2003). On the other hand, we made use of another strain carrying a $P_{GAL1}$-*MAD2-MAD3* construction, which overexpresses a fusion protein of these two key SAC players (Mad2-Mad3[OE]), yielding an active SAC in galactose. Under this condition, the active SAC maintains the mid-M arrest by keeping the APC inactive (Lau & Murray, 2012; Thadani et al, 2018).

We observed a general pattern that was shared by both non-Nz mid-M strategies, and that greatly differed from Nz-arrested cells. Despite an organized bar-like rDNA (Net1-GFP) was seen in all conditions, the horseshoe loop was largely absent and, instead, either a straight bar or a hook that crossed the bud neck orientated in the chromosome division axis (polar axis) was the major outcome (Figs 6B–F and S6C and D). This shift in the rDNA morphology was not a consequence of the newly introduced alleles, as Nz still leads to horseshoe loops in these strains (Fig S6C and E). Importantly, the crossing bar and the hook appear to bend to interact with and wrap around vacuoles in the second body (either the mother or the bud) (Fig 6C, D, and G). The rDNA length was slightly larger in the *cdc16* crossing bars than in the Nz horseshoe loops, pointing out that the rDNA is even more stretched in such configuration (Fig 6E). In the $P_{GAL1}$-*MAD2-MAD3* strain, the bar was shorter (Fig S6F); however, this was due to the incubation with galactose, as Nz-induced horseshoe loops were also shorter in galactose (Fig S6G).

In addition to these two strains, we included a third approach to arrest cells in mid-M, while keeping polymerized microtubules. This was based in a *cin8-aid kip1Δ* strain; Cin8 and Kip1 are partly redundant kinesins required for the assembly of the mitotic spindle (Singh et al, 2018). Unlike the previous strains, which are able to form a bipolar spindle, depletion of both kinesins renders cells with a monopolar spindle (Fig 6A). Thermosensitive *cin8 kip1* double mutants have been shown to arrest in mid-M at the restrictive condition (Hoyt et al, 1992; Roof et al, 1992). Cin8-aid can be efficiently depleted with IAA (Ayra-Plasencia & Machín, 2019), and we accordingly found that this leads to a similar mid-M arrest (Fig S6H). In terms of the relative position of the nucleus, this arrest was intermediate between what was observed for *cdc16* and Mad2-Mad3[OE] and what was seen with Nz, that is, ~45% of cells had a nucleus tightly stretched across the neck (bow-tie phenotype), with the rDNA oriented in the polar axis, and ~35% had the nucleus entirely in one cell body (Figs 6H and S6H). Remarkably, horseshoe loops were scarcely present, even in cells where the nucleus

locates in a single cell body as in Nz (Fig 6B, last bar; and Fig 6H, upper cell).

We conclude that the absence of microtubules is a prerequisite to acquire the horseshoe rDNA loop. However, the mid-M arrest is sufficient to change the rDNA morphology into an organized bar.

## The rDNA protrusion and the nuclear extension depend on active TORC1 and membrane phospholipid synthesis

We have previously shown that conditions that inactivate TORC1 dismantle the rDNA loop (Matos-Perdomo & Machín, 2018). Intriguingly, many reports have demonstrated that TORC1 inactivation activates autophagy, including nucleophagy, thereby promoting the nuclear–vacuolar interaction (Kvam & Goldfarb, 2007; Dawaliby & Mayer, 2010). According to the results shown above, it appears counterintuitive that the horseshoe rDNA loop is absent when the influence of the vacuole on the nucleus should be maximum. To get further insights, we studied the rDNA and nuclear mass morphologies in cells transiting through stationary phase, when TORC1 activity is expected to be low, autophagy high, and most cells appear swollen and with a large vacuole compressing the rest of the cell organelles. In this condition, the rDNA (Net1-GFP) was hyper-compacted, and its structure was barely modified by the vacuole (Fig S7A). Moreover, we could not observe any histone (Hta2-mCherry) handles. The addition of Nz did not change this pattern, demonstrating that Nz only elicits its effects on the rDNA in growing cells, when Nz leads to the mid-M arrest.

Next, we studied the effect of rapamycin addition, a well-known inhibitor of the TORC1 (Heitman et al, 1991; Barbet et al, 1996; Urban et al, 2007). For this, we compared an Nz arrest to both a concomitant Nz plus rapamycin treatment and adding rapamycin to cells previously arrested in Nz. On the two latter, we found that the rDNA morphology was mainly oval without histone handles (Figs 7A and S7B). Alternatively, some mini-loops/handles of Net1/Hta2 (<2 μm) were also visible (Fig 7A, +Nz → rap condition), as we have shown before (Matos-Perdomo & Machín, 2018). We showed above that horseshoe loops and bars are associated with extended nuclei, as seen by rDNA, histone, and NE markers. Thus, we studied nuclear morphology under Nz treatment and TORC1 inhibition by following the NE shape with Sec61-eCFP. In growing cells, most nuclei appear either spherical or oval shaped (Fig S5A and D), except in those cells transiting M phase where the nucleus is stretched along the mother–daughter axis. In such cases, two morphologies are distinguished. The first one is bow-tie shaped, which is shared by cells that are in the late metaphase and early anaphase (Dotiwala et al, 2007). The second one is dumbbell shaped and corresponds to cells in late anaphase. Upon Nz treatment, the NE appears extended, but rarely in the mother–daughter axis (Figs 7B and S5D and E). However, under Nz plus rapamycin the nuclear morphology was mainly spherical/oval again (Fig 7B). Similarly, cells in stationary phase presented a spherical/oval morphology and, once again, the addition of Nz did not change this pattern (Fig S7C).

In a mid-M block, phospholipid synthesis is unabated, and the nuclear membrane expands around the region that contains the nucleolus (Campbell et al, 2006; Witkin et al, 2012). Several lines of evidence pinpoint the Nem1-Spo7/Pah1 complex as a central player

for the control of nuclear membrane expansion. This complex is involved in the balance between membrane phospholipids during growth conditions and lipid droplets during starving conditions; an active Nem1-Spo7/Pah1 complex shifts the balance towards the latter (Siniossoglou et al, 1998; Pascual & Carman, 2013). TORC1 regulates the activity of Nem1-Spo7/Pah1 by keeping Nem1 unphosphorylated and inactive, so that the phospholipid synthesis is favored (Dubots et al, 2014). Accordingly, Nem1-Spo7/Pah1 mutants display nuclear flares and extensions in growing cells (Santos-Rosa et al, 2005; Campbell et al, 2006). For this reason, we studied the phosphorylation status of Nem1 under Nz treatment and TORC1 inhibition. We arrested cells in either Nz alone or Nz followed by rapamycin addition for 1 h, and further compared these conditions with rapamycin treatment in asynchronous cultures (Fig 7C). We observed the same pattern of major phosphorylation shifts that have been described before in asynchronous exponentially growing cells: two bands, P0 and P1, and a third band, P2, after rapamycin addition (Dubots et al, 2014). Strikingly, only the P0 band was seen in Nz. This un(hypo)phosphorylated state suggests a strong Nem1 inhibition in Nz, which was modified only modestly by rapamycin, either after or concomitant to Nz addition (Figs 7C and S7D and E).

Next, we studied the nuclear and rDNA morphologies in cells depleted of Nem1. Instead of using a knockout mutant, we chose a *nem1-aid* allele (Fig S7F), so that we could control when to elicit NE elongation, and thus avoid carryover effects on the nuclear shape during many generations. We found that an overnight culture (~6 generations) with IAA was sufficient for G2/M nuclei in growing cells to exhibit finger-like nucleolar extensions (flares), in agreement with previous reports (Siniossoglou et al, 1998; Witkin et al, 2012). However, this morphology was more fixed than the plethora of morphologies described above for Nz mid-M arrests. In particular, horseshoe loops were seen in less than 20% of G2/M cells and, instead, protruding bars were the main morphological pattern (Figs 7D–F and S7G). Bars and loops were also shorter than in Nz (Fig 7G; mean length of ~2.5 $\mu$m). However, protruding bars turned out to be thicker, and there were examples in which they could be distinguished as hairpin loops (Fig S7G). The position of the cXIIr telomere in the nuclear mass further confirmed that these *nem1* protruding bar are in fact rDNA/cXIIr loops. This indicates that *nem1* forms rDNA loops that are retrained to blossom into horseshoes. This was partly confirmed as Nz addition doubled the presence of horseshoe loops as well as the overall length of loops and bars (Fig 7F and G). Whether open horseshoe or closed hairpin loops, vacuoles were principally juxtaposed to *nem1* flares as well (Fig 7E).

The fact that the rDNA was always found in the nuclear extensions seen in both Nz and *nem1*, raised the question of whether NE tethering of the rDNA was a prerequisite for this phenotype. Thus, we checked both rDNA morphologies and colocalization with nucleoplasm extensions in the *nur1Δ heh1Δ* double mutant, in which rDNA-NE tethering is compromised (Mekhail et al, 2008). We still observed loops and bars (Figs 7J and S7L). The overall proportion of the sum of loops and bars was equivalent to that of the wild type strain, although horseshoe loops were less frequent. They both were found within nuclear extension, including finger-like flares, in 62% of all bars and loops. We did not observe extensions without the rDNA; hence, the rDNA-NE tethering is a prerequisite for neither the presence of nuclear extensions nor the rDNA being in these extensions.

In addition to addressing the effects of an excess in phospholipids for membrane synthesis, we decided to investigate the consequences of a defect in these lipids on the phenotypes described here. It has been shown that mRNAs encoding lipogenic enzymes (Acc1, Fas1 and Fas2), all involved in fatty acid synthesis, increased in G2/M (Blank et al, 2017b). Previous studies have also shown that biosynthesis of fatty acids is necessary for the extension of the nuclear membrane (Witkin et al, 2012; Walters et al, 2014, 2019; Male et al, 2020). Thus, we drew our attention to fatty acid synthesis and its relation to nuclear membrane growth in Nz. We made a Fas1-aid chimera and tested both NE (Sec61-eCFP) and rDNA (Net1-mCherry) morphologies when Fas1 was depleted before Nz addition. Degradation of Fas1-aid in IAA was partial (~66% drop in protein levels; Fig S7H and I); however, this was sufficient to prevent NE extensions upon Nz addition (Fig 7H and I). In these round nuclei, neither horseshoe loops nor protruding bars were observed, with the rDNA mostly seen as extremely short bars and loops (Fig 7H, white arrows). When Fas1 depletion was triggered after the Nz mid-M arrest, most mid-M cells maintained an extended nucleus (Fig 7I, subtracting bow-ties and anaphases), implying that lipid biosynthesis is required to attain the NE expansion in mid-M arrests but not for its maintenance. Finally, we corroborated these findings by using cerulenin, a specific inhibitor of fatty acid biosynthesis (Inokoshi et al, 1994). We treated cells with cerulenin, 1 h before the addition of Nz, which also resulted in spherical nuclei (Sec61-eCFP) and compacted Net1-mCherry signals (Fig S7J). Incidentally, we also observed for both Fas1 depletion and cerulenin more spherical NE morphologies within the bow-tie subgroup (Fig 7I and S7J and K, "bubble bridge"), suggesting a stiffer NE when fatty acid biosynthesis is inhibited.

### The rDNA loop does not depend on known nuclear–vacuolar interactions

From our previous results, the shape of the malleable NE appears to be highly influenced by the stiffer vacuole. When the NE becomes enlarged in mid-M blocks, the vacuole serves as a template on which the extended NE bends around. In this context, the length of the rDNA in protruding bars and horseshoe loops may depend on how intimate the nuclear-vacuole relationship is. For this reason, we decided to study the role of the nucleus-vacuole junctions (NVJs) in the morphology of the rDNA. The NVJs are formed through the formation of Velcro-like interactions between the vacuolar protein Vac8 and the outer nuclear membrane protein Nvj1, which mediate piecemeal microautophagy of the nucleus (Pan et al, 2000; Roberts et al, 2003). Similarly, NE-vacuole contacts are established as sites for lipid droplet biogenesis, which include the proteins Nvj1, Mdm1, Nvj3, Nvj2, and Vac8 (Henne et al, 2015; Hariri et al, 2018). We tagged the Net1-GFP in wild type, double (*mdm1Δ nvj3Δ*) and quadruple (*mdm1Δ nvj1Δ nvj2Δ nvj3Δ*) "Δnvj" mutants, of which, the latter is known to increase the NE–vacuole inter-organelle distance (Hariri et al, 2018). Cells were arrested in Nz for 3 h and the rDNA structure visualized as before. Surprisingly, we still found that most mid-M cells presented an rDNA loop, even in the quadruple Δnvj mutant (Fig 7K). Moreover, when we stained the cells with Blue CMAC

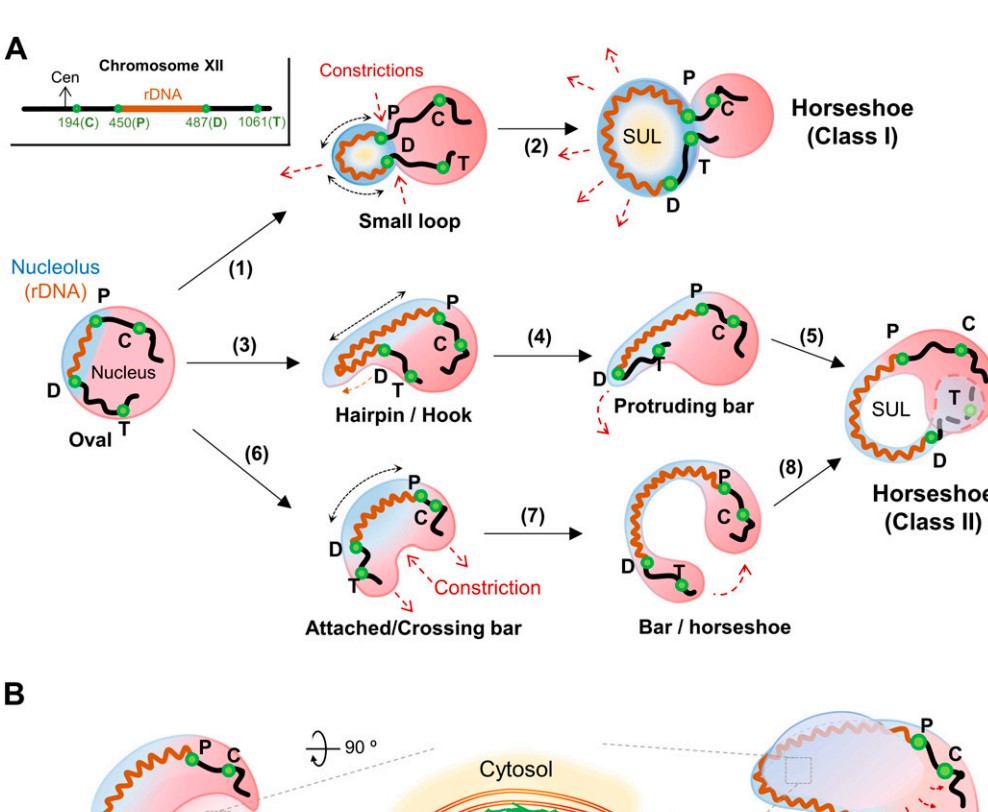

**Figure 8. Classes and origin of ribosomal DNA (rDNA) loops in mid-M arrests.**

**(A)** Relationships between rDNA and nuclear morphologies during the Nz-induced mid-M arrest. Upon Nz addition, the round nucleus expands the NE next to the nucleolus in G2/M cells, so that the rDNA acquires bar-like morphologies distinct to the packed oval shape seen in interphase. Three main reshaping pathways can be drawn. First pathway: (1) the NE grows outwards forming a nuclear bud for the nucleolus, with the corresponding basal constrictions; (2) the NE continues expanding symmetrically in all directions, making possible the flourishing of the class I horseshoe loop, characterized by a unilobed nucleus and a NE ladle at the SUL (yellow). Second pathway: (3) the nucleolus grows outwards as a finger-like projection; (4) its subsequent growth pulls the distal cXIIr arm into the projection; (5) the resulting protruding bar leads to a bi-lobed nucleus that bends until both lobes touch each other, giving rise to the class II horseshoe loop. Third pathway: (6) The NE expands laterally, creating an opposing nuclear constriction; (7) the expanding rDNA bar/handle connects a bi-lobed nucleus; (8) the bar bends and both lobes touch each other, giving rise to the class II horseshoe loop. The rDNA array is depicted as a dark red spring, the rest of the chromosome XII (cXII) as a black thick line, the nucleolus is in blue, and the main nuclear DNA mass is in red. The four cXII regions followed by *tetOs* arrays in this work are indicated as green dots (C, next to the centromere; P, proximal or centromeric rDNA flank; D, distal or telomeric rDNA flank; T, next to telomere; a cXII schematic is also depicted on the left top corner). Black arrows with a number indicate morphological transitions, dashed black double arrows indicate the direction of NE expansion, and red dashed arrows the motion of nuclear subdomains during transitions. **(B)** Models of how the vacuole may serve as a template for the SUL NE ladle. Two different nucleus-vacuole spatial configurations are shown. On the left, the vacuole serves as a scaffold for nuclear reshaping when the NE grows laterally at the nucleolus. This may explain the presence of NE ladles in cashew nuclei (proto-SUL; see Fig 5B). On the right, the expanding NE in a class I horseshoe form an NE ladle at the SUL. Vacuolar engulfment by the NE may explain the ladle shaped of these SULs. At the centre, a zoom-in of the putative NE ladle, showing that it may comprise two very close NE sheets that run in parallel, leaving an extremely narrow nuclear lumen in between. SUL, Space under the rDNA loop; NE, nuclear envelope; NL, nuclear lumen; PNS, perinuclear space; ONM, outer nuclear membrane; INM, inner nuclear membrane.

in the Δnvj mutant, the loops still wrap the vacuole (Fig S7M). We conclude that NE-vacuole contacts are not a prerequisite for the formation of the horseshoe rDNA loop.

### The rDNA loop depends on functional vacuoles

We then checked whether vacuole mutants that impinge on their size, shape, or functionality undermine the ability to form the horseshoe loop. Vacuoles grow and shrink through fusion and fission of vesicles, respectively (Chan & Marshall, 2014). The quick dynamics of such events can be appreciated in the Video 10. Atg18 favors vacuole fission, whereas Apl5, Vps45, and Pep12 are factors involved in delivering vesicles from Golgi to fuse with vacuoles; single knockout mutants for these genes alter vacuolar size and functionality (Becherer et al, 1996; Burd et al, 1997; Cowles et al, 1997; Bryant et al, 1998; Efe et al, 2007). Despite these alterations, we still

observed horseshoe loops and all the other bar-like morphologies in proportions equivalent to the wild-type strain (Figs 7L and S8A).

We next checked the *vac17Δ* mutant, which dampens vacuole maturation and inheritance, and the *vac17Δ pep12Δ* double mutant, which enhances these phenotypes and even make 40% of cells lack normal vacuoles (Jin & Weisman, 2015). We found a clear drop in the proportion of cells with horseshoe loops in the double mutant (Fig 7L; loops in only ~10% of mid-M cells). In addition, the rDNA array in those loops were less organized than in the wild type (Fig S8B).

Last, we checked the effect of suppressing the vacuolar H+-ATPase (V-ATPase), responsible for vacuolar acidification and thus correct vacuolar function (Stevens & Forgac, 1997). We used the *vma1Δ* knockout mutant; Vma1 is one V-ATPase subunit. This mutant grew more slowly than wild type cells and only half of the cell population reached a mid-M arrest after 3 h in Nz. However, in these mid-M cells, the horseshoe loop was barely present, and oval and

small bars were the most frequent outcome instead (Figs 7L and S8C).

### The rDNA loop is not formed when cells are not biochemically in G2/M

Biochemically, the mid-M arrest is characterized by high cyclin dependent kinase (CDK/Cdc28) activity, as well as an inactive Cdc14 (D'Amours & Amon, 2004). Cdc28 and Cdc14 are the master cell cycle kinase and phosphatase, respectively. These biochemical features are shared within a wider cell cycle window, which includes S and G2/M. As shown above in Nz time-course and time-lapse experiments, both rDNA horseshoe loops and nuclear extensions, including the formation of the doughnut-like nucleus, only become visible when cells have a bud large enough to suspect they are in G2/M. Thus, we determined the rDNA morphology in budded cells treated with Nz where molecular conditions differ from this S/G2/M biochemistry. To do so, we forced the unscheduled ectopic release of the Cdc14 phosphatase under Nz treatment. Cdc14 is kept inactive and bound to its inhibitor Net1 in the nucleolus for most of the cell cycle, until its nucleolar release on early anaphase (Shou et al, 1999; Visintin et al, 1999). Upon anaphase onset, Cdc14 inactivates CDK/Cdc28 and dephosphorylates most of its targets, shifting completely the cell cycle biochemistry towards a G1-like state (D'Amours & Amon, 2004). In addition, Cdc14 inhibits transcription by RNA polymerase I, which allows condensin to compact the rDNA (Clemente-Blanco et al, 2009). We constructed a strain carrying a *net1-aid* conditional allele together with Fob1-eCFP and TetR-YFP/*tetO:487* to visualize the rDNA (Net1 binds to Fob1), the nucleoplasm, and the rDNA distal flank, respectively. Degradation of Net1-aid in IAA (Fig S9A and B), allows the ectopic release of Cdc14 outside anaphase, as reported for the *net1-1* allele (Shou et al, 1999). Cells were arrested in Nz or Nz plus IAA and the different rDNA morphologies scrutinized in budded cells. We found bent bars and horseshoe loops under Nz, whereas a rounded nucleoplasm with an oval rDNA on one side was the major morphology in Nz plus IAA (Fig S9C and D). We conclude that the mid-M rDNA loop requires high CDK levels and an inactive Cdc14.

## Discussion

The morphological reorganization of the rDNA in mitosis is one of the most remarkable cytological events in the yeast cell cycle. This is particularly outstanding during mitotic (mid-M) arrests (Guacci et al, 1994; Lavoie et al, 2004; Machín et al, 2005). The nature of such reorganization has deserved multiple studies before, from the early roles of structural maintenance chromosome complexes, mainly condensin and cohesin, to most recent works on polymer–polymer phase separation (Wang et al, 2015; Hult et al, 2017; Lawrimore et al, 2021). Here, we show that growth and deformation of the nuclear envelope (NE) and its interplay with the vacuole plays a major part in the mid-M re-shaping of the rDNA (Fig 8 for a summary of loop classes, putative pathways of loop formation, and interplay with vacuoles). We propose that our findings unify seemingly separated processes and players that influence on the rDNA morphology. On

the one hand, the aforementioned roles of cohesin and condensin and, on the other hand, the selective recruiting of newly synthesized phospholipids to the rDNA-associated NE subdomain in mid-M (Witkin et al, 2012; Walters et al, 2014). From our data, we suggest the latter as the force that reshapes the rDNA until it becomes the outstanding horseshoe loop (Figs 7C–J and 8A). Thus, structural maintenance chromosome complexes may rather provide the foundation to maintain the rDNA organized as an extensible bar during mid-M, while modulating its contraction when required; for example, in anaphase, upon stress, etc. (Matos-Perdomo & Machín, 2019). The extension of rDNA bars would be favored from its spring-like configuration and the presence of locally compacted knotted domains (the observed beads; Fig 2A–D) (Albert et al, 2013; Hult et al, 2017), which could serve as reservoirs to nucleate extension on demand. Thus, the rDNA has the capability to extend for several microns, even when the number of units is relatively low (Figs 2I and J and S2). This would explain why condensin is still needed for the horseshoe loop, which is, nonetheless, the least longitudinally contracted state of the rDNA.

Our findings also shed light on the nature and origin of the outstanding horseshoe rDNA loop, as well as the void space that is left between the loop and the rest of the nuclear mass (SUL), which in turn give rise to the observed doughnut-like nuclei (Figs 1–5 and S1–S5). One of the most striking findings was the fact that there are probably two classes of horseshoe loops in relation to whether the nucleus is uni- or bi-lobed (Fig 8A; class I and II, respectively). It is somehow remarkable that horseshoe loops can originate as convergent morphologies from otherwise different NE reshaping paths that give rise to uni- and bi-lobed doughnut-like nuclei (Fig 8A). These diverse emerging nuclear and rDNA structures suggest a certain degree of randomness when following one of the three pathways we propose (Fig 8A; pathways 1–3), and which are likely influenced by the axis of NE elongation relative to the nucleolus, whether lateral or outwards, and the position, number and size of vacuoles when coming to interact with that nuclear subdomain. On the one hand, the intermediate states we have observed indicate that NE extensions, which result from lipid biosynthesis, can eventually wrap around vacuoles. Either extensive growth of the nucleus or mechanical pressure exerted by vacuoles makes the nucleus become bilobed, with one lobe often carrying most chromosomes, the second lobe the distal part of the chromosome arm that harbors the rDNA array (cXIIr), and the handle that connects both lobes being the rDNA (Fig 8A and B). This spatial configuration fits well with higher order data on chromosome XII organization obtained by chromosome conformation capture (Hi-C), in which regions flaking the rDNA do not interact with each other (Lazar-Stefanita et al, 2017). In this context, the doughnut-like nucleus emerges when the handle bends until acquiring the horseshoe shape and both lobes touch each other. On the other hand, the outward growth of the nucleolar NE pulls the rDNA, so that a horseshoe loop can flourish in the context of a unilobed nucleus (Fig 8A, pathway 1, on the top). In the first scenario, the bi-lobed nucleus with the rDNA horseshoe handle, the non-nuclear nature of the SUL is evident. In the second scenario, the SUL may arise when two opposing sheets of NE get very close to each other so that the nucleoplasm is limited to circulate freely (Fig 8B). In agreement, we found a Sec61 ladle in a significant proportion of

**Table 1. Strains used in this work.**

| Strain[a] | Genotype[b] | Origin |
|---|---|---|
| AS499 | MATa ura3-52 lys2-801 ade2-101 trp1-Δ63 his3-Δ200 leu2-Δ1 bar1-Δ | Strunnikov lab |
| CCG771 | {AS499} NET1-GFP::LEU2 | Aragon lab |
| CCG918 | {AS499} CDC14-GFP::KanMX | Aragon lab |
| CCG1297 | {AS499} TetR-YFP::ADE2; TetO(5.6 Kb)::194Kb-ChrXII::HIS3 | Aragon lab |
| CCG1300 | {AS499} TetR-YFP ADE2; TetO(5.6 Kb)::487Kb-ChrXII::HIS3 | Aragon lab |
| CCG1326 | {CCG1297} NET1-CFP::kanMX4 | Aragon lab |
| CCG1327 | {AS499} TetR-YFP::ADE2; TetO(5.6 Kb)::450Kb-ChrXII::URA3; NET1-CFP::kanMX4 | Aragon lab |
| CCG1328 | {CCG1300} NET1-CFP::kanMX4 | Aragon lab |
| CCG1329 | {AS499} TetR-YFP::ADE2; TetO(5.6 Kb)::1061Kb-ChrXII::HIS3; NET1-CFP::kanMX4 | Aragon lab |
| CCG1582 | {AS499} NET1-CFP::HygB; NUP49-GFP::URA3 | Aragon lab |
| CCG2306 | {AS499} TetR-YFP::ADE2; TetO(5.6 Kb)::450Kb-ChrXII::URA3; TetO(5.6 Kb)::487Kb-ChrXII::HIS3 | Aragon lab |
| CCG2309 | {AS499} TetR-YFP::ADE2; TetO(5.6 Kb)::450Kb-ChrXII::URA3; TetO(5.6 Kb)::1061Kb-ChrXII::HIS3 | Aragon lab |
| CCG2570 (x75) | MATa leu2-3,112 ura3-1 his3-11 trp1-1 ade2-1 can1-100; fob1Δ::his3::HygB (rDNA ~ 75 copies); NET1-GFP::LEU2 | Aragon lab |
| CCG2572 | MATa leu2-3,112 ura3-1 his3-11 trp1-1 ade2-1 can1-100; fob1Δ::his3::HygB (rDNA ~ 25 copies); NET1-GFP::LEU2 | Aragon lab |
| SEY6210 | MATα ura3-52 leu2-3,112 his3-Δ100 trp1-Δ901 lys2-801 suc2-Δ9 | Henne lab |
| SEY6210 mdm1Δ nvj3Δ | {SEY6210} mdm1Δ::KanMX nvj3Δ::NatMX (MATα) | Henne lab |
| SEY6210 ΔNVJ | {SEY6210} nvj1Δ::TRP1 nvj2Δ::HIS3 mdm1Δ::KanMX nvj3Δ:: NatMX (MATα) | Henne lab |
| W303-K699 | MATa trp1-1 can1-100 leu2-3,112, his3-11,15, ura3-1 GAL phi+ ade2-1::OsTir1-9Myc::ADE2 smc4-3HA::TRP1 Net1-yEmCitrine:: HIS3 leu2-3,112::P_{GAL}-MAD2-MAD3::LEU2 | Uhlmann lab |
| yED233 | Mata ura3-1 HTA2-mCherry::URA3 ade2-1 his3-11,15 leu2-3, 112 trp1-1 can1-100 | Pelet lab |
| YNK54 | Mata ura3-1::ADH1-OsTIR1-9Myc::URA3 ade2-1 his3-11,15 leu2-3,112 trp1-1 can1-100 | Kanemaki lab[c] |
| DMY3284- W303a | {W303a} leu2::mURA3 heh1Δ::KANR nur1Δ::HPHR | Moazed Lab |
| FM931 | {YNK54} NET1-GFP::LEU2 | Machín lab |
| FM2113 | {YNK54} cdc14-aid*-9myc::Hph; SPC42-RedStar::KanMX; [NOP1-CFP(LEU2)] | Machín lab |
| FM2301 | {CCG2309}; NET1-eCFP::KanMX4 | This work |
| FM2361 | {CCG1297}; NET1-eCFP:KanMX4 | This work |
| FM2383 | {FM931}; NEM1-6HA::natNT2 | This work |
| FM2394 | {YNK54}; SEC61-eCFP:kanMX4 | This work |
| FM2396 | {FM931}; cdc16-aid*-9myc::hphNT | This work |
| FM2398 | {CCG1300} ura3-52::ADH1-OsTIR1-9Myc::URA3; net1-aid*-9myc::hphNT; FOB1-eCFP::KanMX4 | This work |
| FM2399 | {FM2394}; NET1-eYFP::Hph | This work |
| FM2438 | {CCG2306}; NET1-eCFP:KanMX4 | This work |
| FM2474 | {CCG1297}; cdc15-2:9myc:Hph; cin8::cin8-AID*-9myc::KanMX; ura3-52::ADH1-OsTIR1-9myc::URA3; Δkip1::HIS3MX4 | Machín lab |
| FM2614 | {yED233}; NET1-GFP::LEU2 | This work |

| Strain[a] | Genotype[b] | Origin |
|---|---|---|
| FM2619 | {YAT1735}; NET1-GFP::LEU2 | This work |
| FM2620 | {FM2398} ura3-52 (-OsTIR1) | This work[d] |
| FM2639 | {FM2394}; NET1-mCherry::natNT2; trp1-1::P_ADH1-EGFP-IAA17-NLS::TRP1 | This work[e,f] |
| FM2641 | {FM2394}; NET1-mCherry::natNT2; trp1-1::P_ADH1-EGFP-IAA17::TRP1 | This work[e,f] |
| FM2658 | {CCG2309}; SEC61-eCFP:kanMX4; NET1-mCherry:natNT2 | This work |
| FM2659 | {FM2396}; ura3-1 (-OsTIR1) | This work[d] |
| FM2696 | {SEY6210}; NET1-GFP::LEU2 | This work |
| FM2697 | {SEY6210 mdm1Δ nvj3Δ}; NET1-GFP::LEU2 | This work |
| FM2698 | {SEY6210 ΔNVJ (nvj1Δ,nvj2Δ,mdm1Δ,nvj3Δ)}; NET1-GFP::LEU2 | This work |
| FM2707 | {AS499} TetR-YFP::ADE2; TetO(5.6 Kb)::1061Kb-ChrXII::HIS3; cdc15-2:9myc:Hph; ura3-52::ADH1-OsTIR1-9myc::URA3; HTA2-mCherry:natNT2; NET1-ECFP:klTRP1 | This work |
| FM2735 | {FM2707} cdc16-AID*-9myc:KanMX | This work |
| FM2743 | {yED233}; SEC61-EYFP::kanMX4 | This work |
| FM2748 | {FM2707} nem1-AID*-9myc:KanMX | This work |
| FM2799 | {yED233}; NET1-ECFP::klTRP1; SEC61-EYFP::kanMX4 | This work |
| FM2800 | {FM2639}; fas1-aid*-9myc::hphNT | This work |
| FM2913 | {FM2696}; apl5Δ::HIS3MX6 | This work |
| FM2923 | {FM2696}; atg18Δ::HIS3MX6 | This work |
| FM2924 | {FM2696}; vps45Δ::HIS3MX6 | This work |
| FM2945 | {FM2696}; pep12Δ::HIS3MX6 | This work |
| FM2951 | {FM2696}; vma1Δ::HIS3MX6 | This work |
| FM2959 | {FM2474}; NET1-mCherry::natNT2 | This work |
| FM2960 | {DMY3284-W303a}; trp1-1::P_ADH1-EGFP-IAA17-NLS::TRP1; NET1-mCherry::natNT2 | This work |
| FM2965 | {yED233}; NET1-ECFP::LEU2; VPH1-GFP::kanMX4 | This work |
| FM2973 | {FM2945}; vac17Δ::TRP1 | This work |
| FM2977 | {FM2696}; vac17Δ::TRP1 | This work |

[a]Strains are sorted alphabetically, and then by number, starting from strains reported in previous works.
[b]Curly brackets indicate parental strains used for successive strain construction. Semicolons separate independent transformation events during strain construction; intermediate strains are omitted. Square brackets indicate episomal elements.
[c]This strain was obtained from the NBRP repository (http://yeast.lab.nig.ac.jp/yeast/).
[d]These strains were obtained by counterselecting for the ura⁻ phenotype in five-FOA, which results in the pop out of the OsTIR1::URA3 segment.
[e]These strains express EGFP reporters for the nucleoplasm and the cytoplasm, respectively. These EGFPs are chimeras that contain the auxin-responsive degron peptide IAA17 from *Arabidopsis thaliana*. In most figures, the reference to the IAA17 is omitted for the sake of space because it is experimentally irrelevant. In those cases where cells were treated with IAA, the IAA17 epitope is indicated.
[f]The integrative plasmids for making these strains were also obtained from the NBRP repository (pMK42 and pMK72; originally from Kanemaki lab). To target pop-in integration into the *TRP1* locus, both plasmids were digested with MfeI before transformation.

SULs. Sec61 labels both NE and ER, so we cannot completely rule out that the ladle is formed of ER. Two findings suggest otherwise though; we observed traces of nucleoplasm and Nup49 (NPCs) in a subset of ladles (Figs 5B and S3H).

The horseshoe loop requires nuclear extensions in the absence of microtubules (Nz treatment). When mid-M arrests are accomplished by means that preserve microtubules (APC inactivation, ectopic activation of SAC or monopolar spindles), other rDNA morphologies are seen (Figs 6 and S6). Despite some of them (bars and hooks) can be envisaged as incomplete horseshoe loops because the bipolar spindle pulling forces pose a barrier to vacuole wrapping, non-horseshoe morphologies were seen with a monopolar spindle as well. This points out that the absence of microtubules per se is a requirement for the nuclear reshaping that leads to the horseshoe loop, establishing a connection between the cytoskeleton and the way the nucleus is extended during mid-M arrests.

We also show that membrane phospholipid biosynthesis is required, yet not sufficient, for the horseshoe rDNA loop (Fig 7C–I). It is

likely that this relates to the expansion of the NE surface, as it has been suggested (Siniossoglou, 2009; Walters et al, 2014, 2019; Barbosa et al, 2019; Male et al, 2020). Another layer of control is defined by TORC1 itself (Figs 7A and B and S7A–C), which several studies relate to cell cycle progression (including signaling from mature vacuoles) and G2/M transition (Nakashima et al, 2008; Wang & Proud, 2009; Jin & Weisman, 2015; Moreno-Torres et al, 2017; Pérez-Hidalgo & Moreno, 2017). It could be possible that TORC1 controls the localization and/or activity of specific phospholipid regulators besides the Nem1-Spo7/Pah1 complex and, likely, up-regulation of lipid biosynthesis by TORC1 links the nuclear envelope (expansion) cycle with the chromosome segregation (condensation) cycle, at least in yeast (Tatchell et al, 2011; Barbosa et al, 2015; Sanchez-Alvarez et al, 2015; Takemoto et al, 2016; Blank et al, 2017a, 2017b). Nevertheless, the effects of TORC1 on nuclear and rDNA shape go beyond NE synthesis; TORC1 activity appears essential for both the establishment and maintenance of the horseshoe loop, whereas NE synthesis is only required for its establishment (Fig 7A versus Fig 7I) (Matos-Perdomo & Machín, 2018).

Along with the NE expansion, the presence of vacuoles next to the nucleus greatly influences the final shape of both the nucleus and the rDNA loop. Such influence is especially relevant in the doughnut-like nucleus with the long horseshoe loop, in which a single or multiple vacuoles locate and shape the SUL (Figs 4 and S4). Almost always, vacuoles and nuclear extensions colocalize. Despite the large volume they occupy in the cell predicts such high degree of colocalization, the facts that extensions are juxtaposed to vacuoles since their birth, fold along their surface while they grow, and often leave a hemispheric ladle, strongly suggest that the nucleus-vacuole connection is intimate (Figs 3–5 and S3–S5). This is further strengthened in non-Nz mid-M blocks, in which the presence of microtubules appears to favor the interaction of nuclear extension with vacuoles in the bud (Fig 6). In fact, such interaction may well explain why in these arrests the rDNA locates as a protrusion that crosses the neck and make the nucleus resemble a bow-tie (Palmer et al, 1989; Rai et al, 2017). With that in mind, it is then somehow surprising that horseshoe loops are still present in a quadruple mutant for all known NVJs as well as in mutants that break the rDNA-NE tethering (Figs 7J–L and S7L). Likewise, the effect of mutants that modify vacuole shape, size and number are rather modest (Figs 7L and S8). In the mutants we observed a drop in horseshoes and long bars in favor of smaller and more compacted rDNA shapes (vma1Δ and vac17Δ pep12Δ), a mechanistic connection between these players and TORC1 has been suggested (Jin & Weisman, 2015; Wilms et al, 2017), which points towards a deficient TORC1 activation rather than an isolated effect related to a deficient vacuole homeostasis. Hence, it remains to be determined whether the role of vacuoles in nuclear and rDNA reshaping is active or passive; either way, vacuoles do modulate to a great extent all the observed nuclear and rDNA morphologies.

The doughnut-like phenotypes presented hereinabove, whether based on uni- or bi-lobed nuclei, resembles one atypical nuclear phenotype recently described in human cell lines, the toroidal, or doughnut-shaped nucleus, in which lysosomes (vacuole equivalents in higher eukaryotes) occupy the doughnut hole (Almacellas et al, 2021). This striking morphology results from mitotic errors that

stem from lysosomal impairment. Likewise, whereas we have found a diverse landscape of NE outward extensions during mid-M delays, in an accompanying article, NE inward ingressions have also been found in yeast upon lipid stress (Garcia et al, 2022). In both cases, NE deformations are associated to the nucleolar subdomain and juxtaposed to vacuoles. All these findings underline the malleability of the nucleus and how it is able to acquire extreme shapes far from the ideal sphere. The implications for the functionality and stability of the genome are yet to be determined, but they ought not to be neutral. Finally, in higher eukaryotes, Lipin1 (Pah1 in yeast) is under the control of the mammalian target of rapamycin complex 1 (mTORC1) as well as the biosynthesis of lipids (FASN, ACC; which are orthologs of yeast Fas1, Fas2, and Acc1) (Laplante & Sabatini, 2009; Düvel et al, 2010; Peterson et al, 2011; Menon et al, 2017). The coordination between protein and lipid synthesis is crucial for cell growth (Sanchez-Alvarez et al, 2015; Blank et al, 2017a). Importantly, cancer cells synthesize large amounts of lipids for new membranes and, hence for tumour growth (Bauer et al, 2005; Menendez & Lupu, 2007; Foster, 2009). Not surprisingly, fatty acid synthase inhibitors (FASN inhibitors) are under clinical trials (Falchook et al, 2021). Altogether, our results with yeast cells open promising new avenues for modeling these intricate processes and testing new antitumour drugs in this manageable organism.

## Materials and Methods

### Yeast strains and reagents

Unless noted otherwise, all yeast strains are derivatives of W303 and YPH499 (congenic with S288C). Relevant genotypes of yeast strains used in this study are listed in Table 1. Table S1 lists the correspondence between strains and experiments. Genetic engineering to construct most new strains was carried out through standard PCR-based procedures (Dunham et al, 2015). Table S2 lists the specific reagents used in this study.

### Yeast cell growth and experimental conditions

For experiments, strains were routinely grown overnight in rich YP medium (yeast extract 1% wt/vol plus peptone 2% wt/vol) supplemented with 2% glucose (YPD) at 25°C with moderate orbital shaking (150 rpm; 25 mm orbit). For the standard mid-M arrest, cells were incubated with nocodazole for 180 min. First, nocodazole was added directly to a log phase culture ($OD_{620}$ ~0.8–1.8) at a final concentration of 15 µg/ml, then after 120 min, half the initial concentration was added to the culture media. For aid-based depletion (cdc16-aid, cin8-aid, nem1-aid, fas1-aid, and net1-aid), cells were incubated with indole-3 acetic acid (IAA) at a final concentration of 5 mM (from a 500 mM stock in DMSO). For overexpressing Mad2-Mad3, cells bearing the $P_{GAL1}$-MAD2-MAD3 construction were grown in 2% raffinose and then overexpression from the galactose-inducible GAL1 promoter was accomplished by growing cells in 2% raffinose plus 2% galactose for 240 min. For rapamycin experiments, a final concentration of 200 nM was used (from a 2.2 mM stock in DMSO). DMSO was used at a final concentration of 1%

vol/vol. Cerulenin was added at a final concentration of 2 μg/ml (from a 5 mg/ml stock in EtOH).

### Wide field fluorescence microscopy, staining, and image processing

Two epifluorescence inverted microscopes were used. A Leica DMI6000B with an ultrasensitive DFC350 digital camera was used for single cell visualization with a 63X/1.30 immersion objective as we have reported before (Quevedo et al, 2012; Matos-Perdomo & Machín, 2018). A Zeiss Axio Observer.Z1/7 was also used; this microscope was equipped with an Axiocam 702 sCMOS camera, the Colibri-7 LED excitation system, narrow band filter cubes for co-visualization of CFP, YFP/GFP, and mCherry without emission crosstalk, and a 63X/1.40 immersion objective.

Whenever possible cells were imaged alive. Briefly, 250 μl of cell culture was collected at each time point, centrifuged at 300*g* for 1 min at room temperature, the supernatant carefully retired, and ~1.5 μl of the pellet was added on the microscope slide. Samples were visualized directly using the appropriate filter cube for each tag/stain. For each field, we first captured either single planes or a series of 10–20 z-focal plane images (0.2–0.6 μm depth between each consecutive image), and then we processed images with the Leica AF6000, Zeiss Zen Blue and ImageJ software. For z-stack 2D projections, we used the maximum intensity method. Deconvolution was performed on z-stacks using Leica AF6000 software (method: blind deconvolution algorithm, 10 total iterations, fast processing). Orthogonal projections were generated using ImageJ. Fluorescence intensity profiles were generated using Leica AF6000 and Zeiss Zen 3.1 lite (blue edition) software. For short time-lapse movies of living cells, Nz-blocked cells were pelleted and spread at a high density onto the slide. Specific conditions are described in the video legends.

Stains: DAPI, YOYO-1, SYTO RNASelect, MDY-64, Blue CMAC, carboxy-DCFDA. Nz-blocked cells were stained as follows:

For DAPI staining, the cell pellet was frozen for at least 24 h at −20°C before thawing at room temperature, and then ~1.5 μl of the pellet was added to ~1 μl of 4 μg/ml of DAPI on the microscope slide.

For YOYO-1, we followed a previously described procedure (Ivanova et al, 2020). Briefly, cells were fixed in 4% formaldehyde for 30 min, washed once in PBS, and re-suspended in 5 mg/ml zymolyase in P solution (1.2 M sorbitol, 0.1 M potassium phosphate buffer, pH: 6.2) for 1 min. Cells were spun down, taken up in P-Solution +0.2% Tween 20 + 100 μg/ml RNAse A and incubated for 1 h at 37°C. After digestion, cells were pelleted and taken up in P-Solution containing 25 μM YOYO-1 and visualized as before.

For SYTO RNASelect, cells were pelleted and stained according to manufacturer's procedures. Briefly, a solution of RNASelect green fluorescent stain (final concentration 500 nM) in YPD medium was added to the cells and incubated for 30 min. After this, cells were washed twice with fresh YPD, let rest for 5 min and visualized as before.

For carboxy-DCFDA and Blue CMAC, 1 ml of cells were incubated for ~15–30 min in either carboxy-DCFDA (final concentration of 10 μM from a 10 mM DMSO stock solution) or Blue CMAC (final concentration of 100 μM from a 10 mM DMSO stock solution). Then, the cells were pelleted, washed in YPD (PBS if fixed), and visualized

accordingly. For MDY-64, cells were incubated for ~3–5 min in of 10 μM from a 10 mM DMSO stock solution.

### CSM

Nz-blocked cells were pelleted and imaged in two Zeiss Axio Observer.Z1/7 inverted microscopes equipped for super-resolution confocal microscopy with live cell capabilities (LSM880 with Airyscan and LSM980 with Airyscan 2). The resolution provided in Airyscan mode is lateral (x/y) resolution to 120 nm for 2D and 3D data sets (z-stacks) and 350-nm axial (z) resolution for z-stacks, with an improved resolution up to 1.7X compared with standard confocal (Huff, 2015). The super-resolution images were taken with either a C-Apochromat 63x/NA 1.20 W M27 DICII objective for the LSM880 or a Plan-Apochromat 63x/NA 1.40 Oil M27 DIC objective for the LSM980. The Airyscan detector was used for all single and multiple labellings, with the pinhole automatically set to correct opening according to the selected Airyscan mode. The super-resolution mode of the Airyscan detector was used throughout. We used the following laser lines for excitation of fluorescent tags: 405 nm for CFP; 514 nm for YFP; 488 nm for GFP; and 561 nm for mCherry. The bright field (BF) image was acquired with T-PMT detectors (pinhole 1 AU). After imaging, Airyscan processing was conducted. Z-stack 2D projections were generated by either applying the processing "extended depth of focus" using Zeiss Zen Blue 3.2 software or the sum intensity method in ImageJ.

For 3D reconstructions, ~45–50 z planes (0.15 μm thick) were obtained across the entire cell. Both alive and fixed cells were photographed, although fixed cells were generally used in the presented experiments. For cell fixation, cells were incubated in 3.7% wt/vol formaldehyde on a nutating mixer at room temperature for 15–30 min, spun down at 6,800*g* for 30 s, the pellet washed in 500 μl of filtered PBS 1×, resuspended in another 500 μl of PBS 1×, and stored at 4°C in the dark. Before micrographs were taken, suspensions were sonicated for 8 s in a bath sonicator to separate clumps of cells.

Unless stated otherwise in video captions, long (3 h) time-lapse images were acquired on cells immobilized in Nunc Lab-Tek coverglass eight-wells chambers pretreated with concanavalin A (ConA). ConA pretreatment was undertaken the day of filming by adding 50 μl of ConA (1 mg/ml in PBS) to a well, incubate in the dark at 25°C for 20 min, and washed the well twice with synthetic complete (SC) media. For cell adhesion, a log cell culture was concentrated 2× in SC and 100 μl of the suspension applied to the well and kept at 25°C for 20 min. Non-attached cells were then washed twice with 100 μl of SC media and finally the well was covered with 250 μl of YPD plus Nz at a final concentration of 22.5 μg/ml to start live cell imaging. The higher concentration of Nz used in these experiments is actually equal to the initial plus the reinforcement doses applied in liquid cultures growing in an air incubator. We did this to avoid detachment of the cells by pipetting in the middle of the experiment. Aside from photobleaching and phototoxicity, the horseshoe loop was photosensitive in subtoxic conditions (as determined by comparing end points of exposed versus non-exposed fields). Because of that, laser powers were kept to a minimum and number of z planes reduced. With 405-nm UV irradiation (for CFP), the number of total frames (z planes plus t

points) was set to a maximum of 35; without 405 nm, the maximum frame number was 150.

## TIRF microscopy

This was adapted from a protocol described before (Barroso-González et al, 2009). Nz-blocked cells were pelleted and imaged with an inverted microscope Zeiss 200 M through a 1.45-numerical aperture objective ($\alpha$ Fluar, 100×/1.45; Zeiss). The objective was coupled to the coverslip using an immersion fluid (n(488) 1.518, Zeiss). The expanded beam of an argon ion laser (Lasos; Laser-technik GmbH) was band-pass filtered (488/10 nm) and used to selectively excite EGFP-tagged proteins, for evanescent field illu-mination. The laser beam was focused at an off-axis position in the back focal plane of the objective. Light, after entering the coverslip, underwent total internal reflection as it struck the interface be-tween the glass and the cell at a glancing angle. The images were projected onto a back-illuminated CCD camera (AxioCam MRm; Zeiss) through a dichroic (500 LP) and specific band-pass filter (525/50 nm). Each cell was imaged using Axiovision (version 4.9; Zeiss) with 0.5 s exposure. Image analysis: The raw images were low-pass filtered (3 × 3 pixels) and analyzed with ImageJ.

## TEM

The protocol was adapted and modified from Byers and Goetsch (1991). Cells were arrested in nocodazole for 180 min, then pelleted, re-suspended and fixed in phosphate-magnesium buffered (40 mM $K_2HPO_4$, and 0.5 mM $MgCl_2$, pH 6.5) 2% glutaraldehyde (EM Grade) + 2% formaldehyde and incubated overnight and stored at 4°C. Then, cells were rinsed twice in 0.1 M phosphate-citrate buffer (170 mM $KH_2PO_4$ and 30 mM sodium citrate, pH 5.8) and re-suspended in this buffer containing a 1/10 dilution of Lyticase (10 mg/ml 2,000 U stock) + Zymolyase (5 U/$\mu$l stock) and incubated at 30°C for 2 h, or until cell walls have been removed. For post fixation, cells were washed twice in 0.1 M sodium acetate (pH 6.1), transferred to a 2% osmium tetroxide fixation solution, and incubate for 4 h in a fume hood. Then, cells were washed with double-distilled water (ddH$_2$O) and transfer to 1% aqueous uranyl acetate for 60 min of incubation in the dark. After this, cells were washed twice in ddH$_2$O and de-hydrate by transferring them through a series of ethanol con-centrations (20, 40, 60, 70 [overnight], 96, and 100). Then, cells were pelleted and resuspended in EMBed 812 resin. Finally, semi-thin and ultra-thin sections were cut on an ultramicrotome (Reichert Ultracut S-Leica) and stained with toluidine blue for semi-thin sections and with uranyl acetate and lead salts (Sato's Staining Procedure, 5 min) for ultra-thin sections. EM images were captured by a TEM 100 kV JEOL JEM 1010 electron microscope.

## Pulsed field gel electrophoresis and Southern blot

Yeast chromosomes extraction was prepared in low-melting point agarose plugs as reported before (Ayra-Plasencia et al, 2021). For each sample, six OD$_{600}$ equivalents were centrifuged and washed twice in ice-cold sterile 1× PBS. Then, cells were re-suspended in Lyticase solution (2,500 U/ml), and embedded into 0.5% (wt/vol) agarose plugs. Finally, full-sized chromosomes were obtained by

digesting overnight in RNase A (10 $\mu$g/ml) and Proteinase K (1 mg/ml) containing solutions at 37°C. Pulsed field gel electrophoresis, used to assess the chromosome XII size, was performed by using the CHEF DR-III system (Bio-Rad). One third of each plug was placed within the corresponding well of a 1% (wt/vol) agarose gel made in 1× TBE buffer. Then, the wells were filled-up and sealed with ad-ditional 1% (wt/vol) agarose. 0.5× TBE was used as the running buffer at 14°C. The electrophoresis was carried out at 3 V/cm for 68 h, including 300 and 900 s of initial and switching time (respec-tively), and an angle of 120°. The gel was stained with ethidium bromide for 40 min and destained with ddH$_2$O for 20 min. The chromosomes bands were visualized under UV light using the Gel Doc system (Bio-Rad). To specifically study the chromosome XII, a Southern blot was carried out by a saline downwards trans-ference onto a positively charged nylon membrane (Hybond-N+, Amersham-GE). A DNA probe against the NST1 region within the rDNA was synthesized using the Fluorescein-High Prime kit (Sigma-Aldrich). The fluorescein-labelled probe hybridization was carried out overnight at 68°C. The next day, the membrane was incubated with an anti-fluorescein antibody coupled to alkaline phosphatase (Roche), and the signal was developed using CDP-star (Amersham) as the substrate. The detection was recorded by using the Vilber-Lourmat Fusion Solo S equipment.

## Western blotting

Western blotting was carried out as reported before with minor modifications (Matos-Perdomo & Machín, 2018; Ayra-Plasencia & Machín, 2019). Briefly, 5 ml of the yeast liquid culture was collected to extract total protein using the trichloroacetic acid (TCA) method; cell pellets were fixed in 2 ml of 20% TCA. After centrifugation (2,500$g$ for 3 min), cells were resuspended in fresh 100 $\mu$l 20% TCA and ~200 mg of glass beads were added. After 3 min of breakage in a homogenizer (P000062-PEVO0-A; Precellys Evolution-Bertin In-struments), extra 200 $\mu$l 5% TCA were added to the tubes and ~300 $\mu$l of the mix were collected in new 1.5 ml tubes. Samples were then centrifuged (2,500$g$ for 5 min) and pellets were resuspended in 100 $\mu$l of PAGE Laemmli Sample Buffer (1610747; Bio-Rad) mixed with 50 $\mu$l TE 1X pH 8.0. Finally, tubes were boiled for 3 min at 95°C and pelleted again. Total proteins were quantified with a Qubit 4 Fluorometer (Q33227; Thermo Fisher Scientific). Proteins were re-solved in 7.5% SDS–PAGE gels and transferred to PVFD membranes (PVM020C-099; Pall Corporation). For protein phosphorylation states, we used the method for Phos-tag acrylamide gel electro-phoresis (Kinoshita et al, 2006). The following antibodies were used for immunoblotting: The HA epitope was recognized with a primary mouse monoclonal anti-HA (1:1,000; Sigma-Aldrich); the Myc epi-tope was recognized with a primary mouse monoclonal anti-Myc (1:5,000; Sigma-Aldrich); the Pgk1 protein was recognized with a pri-mary mouse monoclonal anti-Pgk1 (1:5,000; Thermo Fisher Scien-tific) and the aid tag was recognized with a primary mouse monoclonal anti-miniaid (1:500; MBL). A polyclonal goat anti-mouse conjugated to horseradish peroxidase (1:5,000, 1:10,000 or 1:20,000; Promega) was used as secondary antibody. Antibodies were diluted in 5% milk TBST (TBS pH 7.5 plus 0.1% Tween 20). Proteins were detected by using the ECL reagent (RPN2232; GE Healthcare) chemiluminescence method, and visualized in a Vilber-Lourmat

Fusion Solo S chamber. The membrane was finally stained with Ponceau S-solution (PanReac AppliChem) for a loading reference.

## Quantification and statistical analysis

All experiments presented in this study are representative examples, and where stated, three (n = 3) or two (n = 2) independent experiments (biological replicates) are shown. In quantifying experiments involving microscopy, number of cells counted for each condition ranged between 100 and 300, depending on the complexity of the data.

For morphological data, cells were categorized, and the corresponding proportions calculated and represented in bar charts. Where indicated, error bars represent Standard Error of the Mean (mean ± SEM). Unless stated otherwise, only G2/M cells were counted, and these were selected from mononucleated budded cells in which the bud was at least half the size of the mother. In mid-M arrests, the bud size equals that of the mother (dumbbell cell), but we still used the term "G2/M cells" in the y-axis of those charts to make comparisons with experiments where G2/M cells do not necessarily get arrested in mid-M (e.g., Nem1 depletion, rapamycin co-treatments, etc.) Because in dumbbell cells it is difficult to establish which is the mother and which is the bud, we use the term "cell body" to refer to either one.

Quantification of rDNA length and distances was performed with the Leica AF6000 software. These continuous data were represented in box-plots. In these plots, the centre lines depict the medians, box limits indicate the 25th and 75th percentile, and roughly give the 95% confidence intervals for each median. Whiskers represent 5–95 percentile. The mean is shown as "+." Dots represent outliers. *P*-values are represented on each boxplot when comparing two sets of data. For this, assumption of normality was calculated by applying a Shapiro Test. For equality of variances, an F test was used before a *t* test analysis when needed. Unpaired *t* test with Welch's correction was used for the equality of two means. When necessary, a nonparametric Wilcoxon Mann–Whitney U test was applied. Significance level was established at *P* < 0.05 (two tailed). Nonsignificant is denoted by "ns." Statistical analyses were performed with GraphPad Prism (https://www.graphpad.com/) and R software (https://www.r-project.org/).

Cross section profiles of fluorescence intensity were obtained with either the Leica AF6000 or the Zeiss Zen Blue software and represented using GraphPad Prism.

Quantification of bands on Western blots was performed by measuring the intensity of each band in non-saturated conditions using the Bio1D software. The relative amount of aid*-9myc tagged protein (target protein) was estimated using Pgk1 as an internal housekeeping control (loading control protein). Normalization of the target protein relative to the loading control protein was carried out for each lane, and then the fold difference calculated (relative target protein levels) for each lane:

$$Normalized\ density = Target\ protein \times \frac{Loading\ control\ (lane\ 1)}{Loading\ control\ (each\ lane)}.$$

$$Fold\ difference = \frac{Normalized\ density\ (each\ lane)}{Normalized\ density\ (lane\ 1)}.$$

# Supplementary Information

# Acknowledgements

We kindly thank the Luis Aragón, Frank Uhlmann, Mike Henne, Danesh Moazed, Joris Winderickx and Patricia Kane labs for yeast strains and plasmids. We thank María Moriel-Carretero for sharing unpublished results as well as critical reading of the manuscript. We also thank David Machado from the Pharmacology Unit at Universidad de La Laguna for support on TIRF microscopy and José Manuel Pérez Galván from Servicio Investigación Microscopía Avanzada Confocal y Electrónica–SIMACE–at Universidad de Las Palmas de Gran Canaria for support on confocal and electron microscopy. This work was supported by the Spanish Ministry of Science grant BFU2017-83954-R to F Machín. The Agencia Canaria de Investigación, Innovación y Sociedad de la Información supported S Santana-Sosa and S Medina-Suárez through the predoctoral fellowships TESIS2018010034 and TESIS2020010028, respectively

## Author Contributions

E Matos-Perdomo: conceptualization, data curation, formal analysis, investigation, visualization, methodology, and writing—original draft, review, and editing.
S Santana-Sosa: formal analysis, investigation, visualization, and methodology.
J Ayra-Plasencia: formal analysis, investigation, visualization, and methodology.
S Medina-Suárez: methodology.
F Machín: conceptualization, data curation, formal analysis, supervision, funding acquisition, validation, investigation, visualization, methodology, project administration, and writing—original draft, review, and editing.

### Conflict of Interest Statement

The authors declare that they have no conflict of interest.

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
