## [Reviewer comments · Life Science Alliance]

Life Science Alliance

The vacuole shapes the nucleus and the ribosomal DNA loop during mitotic delays

Emiliano Matos-Perdomo, Silvia Santana-Sosa, Jessel Ayra-Plasencia, Sara Medina-Suárez, and Félix Machín

DOI: <https://doi.org/10.26508/lsa.202101161>

Corresponding author(s): Félix Machín, Hospital Universitario Nuestra Señora de Candelaria

Review Timeline:

Submission Date:	2021-07-15
Editorial Decision:	2021-09-21
Revision Received:	2022-06-17
Editorial Decision:	2022-07-20
Revision Received:	2022-07-20
Accepted:	2022-07-20

Scientific Editor: Novella Guidi

Transaction Report:

September 21, 2021

Re: Life Science Alliance manuscript #LSA-2021-01161-T

Dr. Félix Machín
Hospital Universitario Nuestra Señora de Candelaria
Research Unit
Ctra del Rosario, 135
Santa Cruz de Tenerife 38010
Spain

Dear Dr. Machín,

Thank you for submitting your manuscript entitled "Nuclear envelope reshaping around the vacuole determines the morphology of the ribosomal DNA." to Life Science Alliance. The manuscript was assessed by expert reviewers, whose comments are appended to this letter. We invite you to submit a revised manuscript addressing the Reviewer comments.

Thank you for this interesting contribution to Life Science Alliance. We are looking forward to receiving your revised manuscript.

Sincerely,

B. MANUSCRIPT ORGANIZATION AND FORMATTING:

Reviewer #1 (Comments to the Authors (Required)):

Earlier work by several labs established that during a mitotic delay in budding yeast, the nuclear envelope (NE) expands to form a membrane extension, or flare, that includes the nucleolus. In the present manuscript, Matos-Perdomo et al. examine the structure of the rDNA loop within this flare, and they discover that the loop/flare is associated with the nucleolus. Interestingly, the authors show that this association does not rely on the nuclear envelope/vacuolar membrane junctions. They further show that the formation of the NE extension during a mitotic delay requires fatty acid synthesis, and its maintenance requires active TORC1 signaling. This is consistent with published results from the Siniossoglou, Cohen-Fix and they authors labs showing that phospholipid synthesis is required for flare formation. They also show that the extent of the flare occupied by the rDNA is not determined by rDNA copy number. Finally, their TetO fluorescent labeling strategy presents an interesting approach for determining the structure of the rDNA loop in particular, and Chromosome XII in general, within the loop.

While the quality of the experiments is overall high, a significant portion of the results presented in this manuscript recapitulates findings that are already published. For example, Witkin et al. 2012 and Walters et al demonstrated the formation of NE flares/loops that contain the nucleolus, that the space under the flare doesn't contain nucleoplasm or bulk chromatin, and changes in flare as a function of time in the mitotic arrest, ultimately resulting in the loop observed here. They also showed, as was repeated again here, that the flare forms in a pre-anaphase arrest induced by a *cdc16* mutant but not when *Cdc28* is inhibited. Dubot et al. 2014 demonstrated that rapamycin treatment results in reduced levels of phosphorylated Nem1. Finally, Santos-Rosa et al. 2005 and Witkin et al 2012 determined that NE expansion, and the flare in particular, are dependent on phospholipid synthesis. These studies are acknowledged in the manuscript, but the results are shown here, again, nonetheless. If the published results are excluded, one is left with two main findings: That the mitotic flare/loop is associated with the vacuole, and the arrangement of the rDNA within the loop of the flare. As explained below, both observations are interesting but preliminary and will require additional work to establish their significance.

Major comments:

1. The observation that the flare is associated with the vacuole is both interesting and novel. However, at this point it is not clear if the loop shape of the flare requires the presence of the vacuole, or whether the vacuole is an obstacle that the flare must contend with, as the vacuole is often near the nucleolus (Roberts et al, MBoC 2003). Moreover, the mechanism by which the vacuole affects loop formation, if indeed the vacuole plays a role in this process, is unknown. There may be vacuolar mutants that could help distinguish between these two possibilities. Short of that, the authors should use live imaging to further examine this point. This can be easily done using vacuole surface markers (e.g. *Vph1*), which will also circumvent the poor staining by vacuolar dyes. For example, at what point does the flare/loop associate with the vacuole? Using live imaging, Witkin et al showed that the flare starts out as a short knob that extends over time into a loop. Does this NE extension always start at a site where the vacuole and nucleus are juxtaposed? Could it be that the flare recruits the vacuole rather than the other way around? In the absence of live imaging, the authors are overinterpreting their data regarding the role of the vacuole in shaping the loop. They contend that the vacuole actively puts pressure on the extending membrane to bend it into the horseshoe shape, However, without live imaging, it's difficult to assess whether the vacuole actively drives flare bending, as the author claim. Of note, live imaging will also help establish (rather than speculate on) the relationship between the different types of nucleolar structures (see minor comment #6 below).

2. In general, it is difficult to see the association of the NE loop with the vacuole. In fact, in Figure 3 the loop appears to "cross over" the vacuole, and in other instances the loop surrounds a cluster of small vacuoles, but it's not clear if it physically associates with them. Using one or more vacuolar markers, such as *Vph1*-GFP, would greatly improve the quality of the images, not only for live imaging as noted above.

3. The authors attempt to determine the configuration of the rDNA within the loop using a TetR/TetO system to mark different loci along chromosome XII (chr XII), where the rDNA repeats are located and around which the nucleolus assembles. They conclude that the rDNA is threaded through the loop, with one of the chr XII telomeres leading the way and ultimately residing in a second nuclear lobe that forms at the end of the loop. There are two main issues with this interpretation: (a) The use of TetO sites for both chr XII loci, and the position in which the TetO repeats are located, preclude unequivocal determination of rDNA structure, because it is not possible to determine which site is where, and the distances between the loci are such that if a non-

rDNA part of chr XII was in the loop, running anti-parallel to the rDNA, this would be impossible to observe. In this scenario the TetO 450 array would be in the leading edge, with the 1061 locus trailing behind in the bulk chromatin and the chromosome region between 450 and the left telomere (proximal to TetO 194) running through the loop, anti-parallel to the rDNA. This seems especially likely in Figure 3E. It's also curious that the authors often see one of the telomeres at the base of the loop, as they stated, if the ends of the rDNA are there as well. This needs to be resolved. Two different systems for labeling chr XII (for example, the TetO and LacO systems) visualized with binding proteins fused to two different fluorescent proteins, strategically positioning these repeats at additional locations (for example, within the rDNA such that they could distinguish between a lobe at the end of the loop chr XII folded on itself), and following these loci in live cells from the time of the loop's inception will go a long way towards establishing the process by which the rDNA extends throughout the loop. Furthermore, to gain mechanistic insight, and especially if chr XII is led through the loop by one of its telomeres, the authors could probe whether this is dependent on telomere-NE attachments, as described by the Gasser and other labs. (b) If indeed there is a lobe at the end of the loop, one could visualize it by confocal microscopy and 3D rendering, as was done previously (see Webster et al JCB 2010).

4. The NE expand in the region adjacent to the nucleolus under conditions of Pah1 inactivation (e.g. *spo7Δ*, *pah1Δ* etc). Examining whether the *spo7Δ* flare, for example, is also associated with the vacuole could add novel content to this study. Either outcome (namely associated or not associated) would be interesting.

5. Another observation that can be expanded to enhance the novelty of the study is that the addition of rapamycin causes the loop to disappear. It's implied (but not shown directly, which is an issue because rapamycin does many things) that this is due to lack of phospholipid synthesis. That continuous phospholipid synthesis is needed to maintain the loop is novel and interesting: what happens to the loop when phospholipid synthesis is blocked? Specifically- does the membrane get reabsorbed into the nucleus, leading to a large nuclear surface area? And why is there a need to continuous phospholipid synthesis once the loop has already formed?

6. The authors conclude that "the absence of microtubules is a prerequisite to acquire the horseshoe rDNA loop that wraps around the vacuole". An alternative possibility is not that microtubule per se inhibit loop formation, but that in a mid-mitosis arrest in the presence of microtubules, the nucleus traverses the neck, and as a result that nucleolus no longer has access to the vacuole. Therefore, what is critically missing in this analysis is visualization of the rest of the nucleus, to determine if, under these conditions, the nucleolar region still extends away for the rest of the nucleus. If it does, but it is not associated with the vacuole, this raises questions regarding the physiological relevance of the loop-vacuole association. Specifically, what would be the physiological conditions that would require loop-vacuole interactions only where there are no microtubules?

Minor issues:

1. The title does not reflect the findings in the study, and in particular the circumstances under which the NE forms the flare/loop in budding yeast. Moreover, as indicated above, the current title is an interpretation that is not based on conclusive evidence. A more appropriate title would be something along the lines of "Nuclear envelope expansion during a mitotic delay in budding yeast is associated with the vacuole". That the rDNA is in this loop was already shown as is not novel to this study.
2. First paragraph of the results: Walters et al 2014 measured the nuclear/cell volume ratio of cells with mitotic flares/loops and found it to be the same as in cells without a loop.
3. The observation that they mitotic flare/loop is independent of the number of rDNA copies is consistent with Walters et al 2014, who showed that the loop forms even where there are no rDNA repeats.
4. Sections recapitulating published data (see above, for example Figure 2A through K, Figure 4, Figure 6F and possibly other Figures/panels) should be removed.
5. The values for non-rDNA region compaction are meaningless because the path of the DNA cannot be traced and the assumption of a straight line is unfounded.
6. Figure 3: could the different shapes of the rDNA reflect how long these cells were arrested in nocodazole? See Witkin et al 2012 for changes in rDNA shape over time as cells arrest in nocodazole.
7. The stationary phase experiment is not informative in the context it is presented: it has already been established that in cycling cells the flare/loop is specific to mitosis (here and mostly in published data), and in stationary phase the nucleus adopts a completely different form. Without delving into that just showing the nucleolus does add much.
8. The discussion contains sections related to rDNA condensation which is out of place for this study, as no experiments in this regard were presented.
9. Please number all pages- it will make it much easier to comment on specific statements, not to mention easier to read when printed.

Reviewer #2 (Comments to the Authors (Required)):

This paper by Matos-Perdomo et al. explores the nature of a previously described observation that ribosomal DNA (rDNA) becomes extended and forms a "horseshoe-like loop" upon treatment with nocodazole (Nz). This paper is an in depth and rigorous exploration of the biogenesis of these structures. They find that the length of the rDNA loop does not depend on the number of repeating rDNA subunits. They use superresolution microscopy of tetO arrays to demonstrate that the loop is comprised of the entire length of two sister chromatids with their telomeres flanking both ends of the loop. They then show that

the rDNA loop wraps around the vacuole, which requires depolymerized microtubules, active TORC1, and phospholipid synthesis. Interestingly, the loop still wrapped around the vacuole when the only established membrane tethering complex between the nucleus and the vacuole was abolished. Finally, they show that the formation of the rDNA loop requires cells to be in the G2/M stage of the cell cycle. Overall, the quality of the data is good, and the conclusions are all appropriate and consistent with the data. I suggest only a few minor experiments, which are listed below:

- 1) The observation that the formation of rDNA loops requires phospholipid synthesis suggests that the aberrant synthesis of membrane adjacent to the nucleolus is responsible for the extended structure the rDNA adopts. It remains to be shown, then, whether the rDNA plays any role in the formation of the loops. I suggest disrupting the association of the rDNA to the inner nuclear membrane by deleting the CLIP complex (Heh1 and Nur1) and determining if the loop still forms and whether or not the rDNA is in the loop.
- 2) The TEM images presented throughout the paper (in particular Figure 2J) are much too small to make out any detail and should be made larger so that the membranes can be more easily visualized.
- 3) "TORC1 complex" is redundant. TORC1 stands for target of rapamycin complex 1 so "complex" can be removed.

Reviewer #3 (Comments to the Authors (Required)):

Matos-Perdomo et. al., report interesting distinct morphologies of rDNA loop that adopts during the mid-M arrests. The rDNA loop results in the appearance of a space under the loop (SUL) which is devoid of any nuclear component but colocalizes with the vacuole. Authors show that the formation and maintenance of the rDNA loop and the SUL require TORC1 and membrane lipid synthesis but its independent of nuclear-vacuolar junctions. While the rDNA loops are unique, the observations shown here lack explanation in several instances. Please find below major and minor comments that authors need to address.

Major comments:

Is there any pre-existing evidence for rDNA loops to get unwound and separate in two sister strands by undergoing threading and knotting? Is this process result of break and repair by reannealing?

Are the constrictions in the CSM images potential sites for sister chromatids to break and repair?

In general, Sec61 staining is not convincing, how do you differentiate external and internal faces of rDNA loops?

Fig 2G, 2H, 2I, 2K are not convincing. Better representative images are required. Also, include graphs of scanning intensity on a line across the image to indicate co-localization.

Fig 2J, 3G- highlight the double membrane.

Please confirm nuclear flares in Fig 3. using nuclear protein PUS1.

Can you test the role for dismantling of spindle apparatus in rDNA horseshoe loop formation without microtubule depolymerization, may be spindle keeps the NE from extending and bending?

Interestingly, the rDNA length was slightly larger in the straight line than in the horseshoe loop, pointing out that the rDNA is even more stretched in such configuration (Figure 4G). However, in galactose, a condition needed to active the SAC in the PGAL1- MAD2-MAD3 strain, the line was shorter (Figure 4H) --- little too much of interpretation. what is n used for counting this? Nz+rapamycin -

Can you separate the effects of the absence of MT from inhibition of TORC? Add rapamycin first and then nz; so even though TORC is absent, mt is disassembled. Is the concentration the same for nz and rapamycin concurrent treatment vs step by step?

What happens to rDNA loops in mid-M arrested nem1 mutants?

Figure 5I- For partial FAS-aid degradation experiment authors show that rDNA looping is not affected but only NE extension is affected. Please explain this result. Isn't there a NE around the rDNA handles visible in the representative image?

Fig 6E- first two bars are explained in the results, what is the third bar showing? It seems like that condition is rescuing the defect. Need explanation.

Fig 6F - the rDNA signal is still very close to the vacuolar boundary even though you don't see the extension. Please explain.

The authors should also discuss more about the sequence of events. NE extension providing space for the rDNA knots/beads to get accommodated or the rDNA loop is a cause for forming more NE around them.

Is vacuolar morphology and function important for this? Authors can check by using vps mutants defective in vacuolar fusion and vma mutants defective in V-ATPase function.

Minor comments:

If NVJ is not important for this interaction, it's interesting to still see the rDNA cluster very close to the vacuole. What keeps it close to the vacuole. Please explain.

Authors mention "this chapter" in the mss several times. I think it is not suitable for a research article.

I would suggest rearranging some data from main figures to supplementary figures. especially the cdc14/cdk28 data is not that strong at this point so perhaps can be moved to supplement.

Rebuttal Letter to manuscript LSA-2021-01161-T

First, we would like to thank all reviewers for their comments to improve our work and apologize for the time it has taken us to reply. For various reasons, we have had several delays in the period that we initially had in mind. An important reason was that during the process of sending and reviewing the original MS, our institution acquired a new Zeiss microscope (LSM980), equipped for both confocal super-resolution microscopy (CSM), using Airyscan 2 technology, and wide field fluorescence microscopy (WFM), including the newest Colibri 7 excitation system. We thought it was worth waiting until having the new microscope installed and running to undertake some of the key experiments requested by the reviewers; in particular, the 3D reconstructions of nucleus/rDNA/vacuoles and the movies of loop formation in Nz. Unfortunately, and in addition to the necessary of period to get used to the new microscope, we had a series of initial setbacks with the installation, operation, and configuration of the equipment. This forced us to postpone experiments until all issues were sorted out. We believe, however, that this new microscope has allowed us to add new material that greatly improves the quality of our work. We hope the reviewers will agree with that.

Having said that, we would like to give an overview of the revised manuscript before moving on to respond to the reviewers' specific points. In general, all three reviewers agreed that they would have liked to know more about both the spatial organization of rDNA and the nucleus in mid-M arrests (not only Nz but other arrests that preserve microtubules), as well as the influence of the vacuole on mid-M arrests. I believe that we now provide new data that (i) endorse the bulk of our initial conclusions (on the horseshoe loop, the nature of the SUL, the role of the vacuole, the reorganization of the nucleus towards bilobed/doughnut-shaped morphologies, the role of membrane synthesis, etc.), and (ii) provide new knowledge to our initial study. In relation to the latter, we would like to highlight the presence of a NE/ER ladle in the SUL, detected and characterized from 3D reconstructions, as well as the organization of the rDNA and the flanking regions in all the structures that we observe. Finally, it is worth mentioning the new data obtained through time course experiments and live imaging, which allow us to provide a general, and relatively complex, model by which the horseshoe loop is formed as a convergent morphology from various distinctive changes in the shape of the nucleus in mid-M arrests (new Fig 8 for schematics).

Finally, just mention that the revision contains substantial new material in the form of new panels in main and supplementary figures, as well as up to ten movies (the journal's limit) with several nuclei each. The new data includes novel or improved prototypical morphological examples, quantifications, new charts with a greater number of repetitions, new diagrams, etc. The original version had been prepared to fit the strict requirements of the journal to which the article was submitted first (JCB). In LSA, these requirements are laxer, so we have added more main figures in the revision. These figures come from both the new data and the division of very dense figures in the original version. We have tried to maintain the thematic coherence of each figure as much as possible. The order of some results has been modified accordingly. The table below outlines all changes and the correspondence between old and new figures.

Our replies to reviewers' specific comments are in blue text beneath the corresponding paragraph. While trying to be concise in the answers, we also include notes at the end of the document with extended information on some of our viewpoints. References to the first version are indicated as "MS-v0", whereas the revised version is denoted as "MS-R1".

Table of correspondence between old and new figures

New Fig / Panel	A	B	C	D	E	F	G	H	I	J	K	L	M
Fig 1	1A	new	new	new	new	new	---	---	---	---	---	---	---
Fig 2	1D	1E	1F	1G	1I	1J	new	3C	1B	1C	---	---	---
Fig 3	S2A	2G*	new	2H*	2I*	2J	new	3J*	new	---	---	---	---
Fig 4	2K*	2L	3B	new	new	new	new	new	2M	---	---	---	---
Fig 5	new	new	new	new	new	3D	new	new	new	---	---	---	---
Fig 6	4A	4B + new	new	new	4G	4E	4F	new	---	---	---	---	---
Fig 7	5B + S5D	5F	5G	new	new	new	new	5I	5J + new	new	6B	new	
Fig 8	new	new	---	---	---	---	---	---	---	---	---	---	---
Fig S1	S2B	S2C	S2D	---	---	---	---	---	---	---	---	---	---
Fig S2	S1A	S1B	S1C	---	---	---	---	---	---	---	---	---	---
Fig S3	2A	2B	2C	2D	2E	new	S3A	new	2J	S4B	new	---	---
Fig S4	new	S3D	S3E	S3F	---	---	---	---	---	---	---	---	---
Fig S5	3K	3G	3J	5D + S5D	S4D	S4E	S4A	---	---	---	---	---	---
Fig S6	S5A	S5B	4C	new	S5C	4H	new	new	---	---	---	---	---
Fig S7	5A	5C	S5F	5H	S5G	new	new	S5H	S5I	5K	S5J	new	6C
Fig S8	new	new	new	---	---	---	---	---	---	---	---	---	---
Fig S9	S6A	S6B	6D	6E	---	---	---	---	---	---	---	---	---
Fig S10	S6C	S6D	S6F	6F	---	---	---	---	---	---	---	---	---
movie 1	movie 1												
movie 2	movie 2												
movie 3	movie 3												
movie 4	movie 4												
movie 5	new												
movie 6	new												
movie 7	new												
movie 8	new												
movie 9	new												
movie 10	new												
Table 1	Table S1 (list of strains)												
Table S1	new (strains and experiment type in each figure panel)												
Table S2	new (reactives)												

new: new data in MS-R1; *: new ROI in MS-R1 (mostly CSM images that replaces old WFM images)

Reviewer #1 (Comments to the Authors (Required)):

Earlier work by several labs established that during a mitotic delay in budding yeast, the nuclear envelope (NE) expands to form a membrane extension, or flare, that includes the nucleolus. In the present manuscript, Matos-Perdomo et al. examine the structure of the rDNA loop within this flare, and they discover that the loop/flare is associated with the nucleolus. Interestingly, the authors show that this association does not rely on the nuclear envelope/vacuolar membrane junctions. They further show that the formation of the NE extension during a mitotic delay requires fatty acid synthesis, and its maintenance requires active TORC1 signaling. This is consistent with published results from the Siniosoglou, Cohen-Fix and their authors labs showing that phospholipid synthesis is required for flare formation. They also show that the extent of the flare occupied by the rDNA is not determined by rDNA copy number. Finally, their TetO fluorescent labeling strategy presents an interesting approach for determining the structure of the rDNA loop in particular, and Chromosome XII in general, within the loop.

While the quality of the experiments is overall high, a significant portion of the results presented in this manuscript recapitulates findings that are already published. For example, Witkin et al. 2012 and Walters et al demonstrated the formation of NE flares/loops that contain the nucleolus, that the space under the flare doesn't contain nucleoplasm or bulk chromatin, and changes in flare as a function of time in the mitotic arrest, ultimately resulting in the loop observed here. They also showed, as was repeated again here, that the flare forms in a pre-anaphase arrest induced by a *cdc16* mutant but not when *Cdc28* is inhibited. Dubot et al. 2014 demonstrated that rapamycin treatment results in reduced levels of phosphorylated Nem1. Finally, Santos-Rosa et al. 2005 and Witkin et al 2012 determined that NE expansion, and the flare in particular, are dependent on phospholipid synthesis. These studies are acknowledged in the manuscript, but the results are shown here, again, nonetheless. If the published results are excluded, one is left with two main findings: That the mitotic flare/loop is associated with the vacuole, and the arrangement of the rDNA within the loop of the flare. As explained below, both observations are interesting but preliminary and will require additional work to establish their significance.

It is true that our findings and those the reviewer refers to match very well in the end; however, we would like to stress out that our study revolved around the rDNA loop, a prototypical example of higher-order organization of chromosomes, whereas the other studies dealt with nuclear envelope (NE) extension. We could say that our study is rDNA-centric and the others are NE-centric. Thus, the main objective of the work presented in this manuscript was to study the mid-M rDNA loop and the nature of the Space Under the Loop (SUL).

Previous works, among them Walters et al, demonstrated that a nuclear extension ("flare") forms adjacent to the nucleolus. Although it can be deduced that there is a SUL from their works, this is also true for other much earlier works, including ours, which looked by either FISH or Net1-GFP at the rDNA loop and the DAPI signal under nocodazole arrest (Guacci *et al*, 1994; Lavoie *et al*, 2000, 2002; Fuchs & Loidl, 2004; Machín *et al*, 2005). Besides, the SUL was neither mentioned nor studied in any previous works. In many instances that SUL appeared "open"; i.e., a highly bent nuclear flare. However, in most examples we showed in MS-v0, it was a "closed" SUL. As we now show in MS-R1, whether the SUL is open or closed depends a lot on (i) the spatial perspective the rDNA loop is seen in 2D projections, and (ii) the presence of flanking rDNA regions in the loop (the latter was already mentioned in MS-v0). There are "open" SULs indeed (left by protruding rDNA bars), but they are less frequent than they seem at first glance. Thus, it is sensible to assume that closed SULs might encompass just a specialized and enlarged nucleolus, with the rDNA at the periphery and all rRNAs and pre-ribosomes in the centre, and so we initially thought¹. In the end, the closed SUL has turned out not to be nuclear, but we hope the reviewer understands our initial point of view and why we chose to determine the nature of that SUL. Only after realizing it was not nuclear, we searched for its alternative nature and found the vacuole, and also found that both the protruding rDNA bars and the spectacular horseshoe rDNA loop were intimately surrounded by the NE, establishing then the connection with the work in Cohen-Fix's lab. Having said that, just realize that the NE "flare" they proposed (as a finger-like or knob projection that grows outward from the nucleolus) is only one of the pathways of NE reshaping that explain the origin and the final shape of the loop and other observed nuclear/rDNA morphologies (we now show this better in MS-R1; see below for new data).

As for comments on the other data in MS-v0, we would like to point out they are not mere recapitulations of previous experiments; there is a clear purpose for them in this work. As the reviewer points out, a *cdc16* mutant induces the flare, and we obviously agree with this, but this was not the purpose of experiments in MS-v0 Figure 4 (Figure 6 in MS-R1). Firstly, flares were shown before to be restricted to specific mid-M arrests and to null mutants of the Nem1/Spo7 Pah1 complex; but does this necessarily imply that flares lead to an rDNA loop? In other words, some flares may not form a horseshoe loop in the end, as is the case in non-Nz mid-M arrests. In MS-R1, we now explain better this underlying rationale while providing new data with an improved *cdc16-aid* strain (more labelled markers), and further show how the ability to interact with vacuoles in different cell bodies may account for the differences between the phenotypes observed in Nz and *cdc16* (Figs 6 and S6 in MS-R1). Secondly, the MS-v0 Figure 4 showed means to achieve mid-M arrests in which different variables can be separated, leading to the conclusion that the lack of microtubules greatly enhances the mid-M rDNA horseshoe loop. We further extend on this in the reply to major point 6 and provide new data and strains (monopolar vs bipolar spindle) that greatly reinforced this assessment (Figs 6B, 6H and S6H in MS-R1). Finally, we have now included what the rDNA shape looks like in *nem1* flares before and after adding Nz; note that *nem1* flares are not like the rDNA horseshoe loops seen in Nz (Fig 7D-G in MS-R1).

Finally, (Dubots *et al*, 2014) (please see specially Fig3 and Fig4 of their work) showed that rapamycin results in the phosphorylation of Nem1, contrary to what the reviewer claims (probably a misunderstanding) that rapamycin results in reduced levels of phospho-Nem1. Nem1 is underphosphorylated and inactive when is inhibited by TORC1 (growth conditions; rich media) and becomes hyperphosphorylated and active (more species and increased levels of phospho-Nem1) when cells are treated with rapamycin. Taking this into account, the purpose of the experiment was not to show that rapamycin treatment results in the appearance of more phospho species as done before, but to associate the role of TORC1 with the phospholipid metabolism (Nem1/Spo7 Pah1 complex) and the formation of nuclear extensions and the rDNA loop in Nz, as we had already showed (Figure 4F-H and S5D, E & G in MS-v0). In this way, we further link two important processes for the rDNA loop formation, namely the activity of the TORC1 (other roles for TORC1 on the rDNA loop will be explained in the reply to Major point 5) and the phospholipid metabolism (Nakashima *et al*, 2008; Wang & Proud, 2009; Blank *et al*, 2017). Incidentally, we are not aware that the effect of rapamycin on the Nem1/Spo7 Pah1 complex had been addressed before in mid-M cells.

Major comments:

1. The observation that the flare is associated with the vacuole is both interesting and novel. However, at this point it is not clear if the loop shape of the flare requires the presence of the vacuole, or whether the vacuole is an obstacle that the flare must contend with, as the vacuole is often near the nucleolus (Roberts *et al*, MBoC 2003). Moreover, the mechanism by which the vacuole affects loop formation, if indeed the vacuole plays a role in this process, is unknown. There may be vacuolar mutants that could help distinguish between these two possibilities. Short of that, the authors should use live imaging to further examine this point. This can be easily done using vacuole surface markers (e.g. Vph1), which will also circumvent the poor staining by vacuolar dyes. For example, at what point does the flare/loop associate with the vacuole? Using live imaging, Witkin *et al* showed that the flare starts out as a short knob that extends over time into a loop. Does this NE extension always start at a site where the vacuole and nucleus are juxtaposed? Could it be that the flare recruits the vacuole rather than the other way around? In the absence of live imaging, the authors are overinterpreting their data regarding the role of the vacuole in shaping the loop. They contend that the vacuole actively puts pressure on the extending membrane to bend it into the horseshoe shape, However, without live imaging, it's difficult to assess whether the vacuole actively drives flare bending, as the author claim. Of note, live imaging will also help establish (rather than speculate on) the relationship between the different types of nucleolar structures (see minor comment #6 below).

There are several separate requests within this paragraph:

1.1. As for the vacuole mutants, we assume the reviewer referred to mutants that affect vacuole physiology, shape or abundance; vacuoles are essential for growth and, as far as we know, there are no mutants that render cells without vacuoles. On the other hand, the mutants for vacuole-nuclear junctions were already studied in MS-v0. Regarding ways to alter vacuole

physiology, shape and/or abundance, we had already included two basic approaches: stationary phase and *cdc28* (note that we have opted to remove *cdc28-aid* in MS-R1; see reply to reviewer #3). Both strategies render cells with very large vacuoles, yet they differ in how active TORC1 is (as an active TORC1 appears essential to obtain the rDNA loop in mid-M arrest). In neither case we observed loops, but cells were not in mid-M. We acknowledge this was a drawback. What we have now done is to test the presence of the loop in several non-essential mutants that affect vacuolar inheritance, physiology, and morphology (see also reply to reviewer #3)². So far, the loop was still present in most mutants but *vma1Δ* and *pep12Δ vac17Δ* (Figs 7L and S8 in MS-R1). The latter two suggests that proper vacuole physiology and inheritance is important for the establishment/maintenance of the rDNA loop. We include these new data, but we also discuss whether these links are direct or indirect as vacuole mutants are also expected to affect the TORC1 pathway (Jin & Weisman, 2015; Wilms *et al*, 2017).

1.2. As requested, we have performed experiments to answer whether the “flare” starts at the nucleus-vacuole junction (NVJ) or the vacuole is recruited later on. Note, however, that we now provide new data suggesting that not all nuclear extensions can be considered as “flares” (finger-like NE projections that grow outwards from the nucleolus). In fact, “flares”, or at least what we understand as “flares”, corresponds to just one out of three major nuclear extensions seen after Nz addition (Fig 8 in MS-R1). To address whether vacuoles interact with rDNA-containing nuclear extensions, we used several approaches, though many were non-informative³, so we ended up focusing on Nz time-course experiments and live imaging⁴ with NE/nucleoplasm/rDNA markers vs vacuolar markers (Vph1 for live imaging). We think that we now demonstrate that most extensions touch vacuoles, and this happens from the beginning (Figs 5D,E and movie 10 in MS-R1).

1.3. As for the relationship between bars/flares and horseshoe loops, we now demonstrate that the situation is much more complex and variable than we had anticipated in MS-v0. 3D reconstructions, movies and time-course experiments now showed that there are two major classes of rDNA loops, and that the horseshoe loop is a convergent morphology that stems from different origins when the nucleus reshapes after Nz addition. In addition, we now show that flares can contain a “closed” rDNA loops, which may appear as a leaning flare/protruding bar in lateral views, but that eventually either open ups from the centre (“flourish”), forming a mature horseshoe loop in a unilobed nucleus, or recoils the flanking rDNA regions to later form horseshoe loops in the context of a bilobed nucleus. This conclusion has been possible after individually analysing the set of *tetOs* we have in combination with the rDNA, the nucleoplasm and the NE, as well as precise 3D reconstructions by Zeiss’ Airyscan 2 technology in a LSM980 (Figs 3, 5, 8, S3, S5 and movies 5-9 in MS-R1). The extensive use of confocal superresolution microscopy (CSM) has resulted in a major quality improvement of the data we had previously obtained by epifluorescence wide field microscopy (WFM).

2. In general, it is difficult to see the association of the NE loop with the vacuole. In fact, in Figure 3 the loop appears to “cross over” the vacuole, and in other instances the loop surrounds a cluster of small vacuoles, but it’s not clear if it physically associates with them. Using one or more vacuolar markers, such as Vph1-GFP, would greatly improve the quality of the images, not only for live imaging as noted above.

Those examples of loops that “cross over” vacuoles were 2D projections of the loop and the vacuole. Even in those 2D flattened images, it was often evident that the vacuole sits on slightly lateral views of the horseshoe loop. The visual appreciation of loop-vacuole relationship is largely determined by the spatial orientation of the loop and the vacuoles in those 2D examples. We have now provided better examples in which this complexity can be appreciated (Fig S4A, 5E and movie 10 in MS-R1) as well as a 3D example of a rDNA loop leaning on a larger vacuole (labelled with Vph1, as suggested) (Fig 4G, H in MS-R1). In that example we show that the centre plane of the vacuole and the loop are in different z planes, whereas the loop touches the vacuole at the bottom (or top, depending on the point of view). Note, however, that vacuolar lumen markers (such as Blue CMAC) are often superior to VM markers as the latter give the better intensity and resolution at the centre plane of the sphere (vacuole), whereas the former labels all vacuolar content as a ball.

3. The authors attempt to determine the configuration of the rDNA within the loop using a TetR/TetO system to mark different loci along chromosome XII (chr XII), where the rDNA

repeats are located and around which the nucleolus assembles. They conclude that the rDNA is threaded through the loop, with one of the chr XII telomeres leading the way and ultimately residing in a second nuclear lobe that forms at the end of the loop. There are two main issues with this interpretation: (a) The use of TetO sites for both chr XII loci, and the position in which the TetO repeats are located, preclude unequivocal determination of rDNA structure, because it is not possible to determine which site is where, and the distances between the loci are such that if a non-rDNA part of chr XII was in the loop, running anti-parallel to the rDNA, this would be impossible to observe. In this scenario the TetO 450 array would be in the leading edge, with the 1061 locus trailing behind in the bulk chromatin and the chromosome region between 450 and the left telomere (proximal to TetO 194) running through the loop, anti-parallel to the rDNA. This seems especially likely in Figure 3E. It's also curious that the authors often see one of the telomeres at the base of the loop, as they stated, if the ends of the rDNA are there as well. This needs to be resolved. Two different systems for labeling chr XII (for example, the TetO and LacO systems) visualized with binding proteins fused to two different fluorescent proteins, strategically positioning these repeats at additional locations (for example, within the rDNA such that they could distinguish between a lobe at the end of the loop chr XII folded on itself), and following these loci in live cells from the time of the loop's inception will go a long way towards establishing the process by which the rDNA extends throughout the loop. Furthermore, to gain mechanistic insight, and especially if chr XII is led through the loop by one of its telomeres, the authors could probe whether this is dependent on telomere-NE attachments, as described by the Gasser and other labs. (b) If indeed there is a lobe at the end of the loop, one could visualize it by confocal microscopy and 3D rendering, as was done previously (see Webster et al JCB 2010).

Again, there are several separate requests within this paragraph:

3.1. The spatial configuration of the rDNA horseshoe loop does not include events where the rDNA array folds on itself. This was already tested and shown in Figures 1I & J in MS-v0 (Fig 2E & F in MS-R1). Nonetheless, there are some seeming protruding bars that are actually "closed" loops, which could not be easily resolved in 2D projections. In Nz, these closed loops were rare (<5% of all morphologies) but were more abundant upon Nem1 depletion (without Nz co-treatment) (Figs 2G, 5G, 7D-G & S7G in MS-R1).

3.2. We have now addressed the configuration of each *tetO* individually⁵ (Fig 5G-I in MS-R1), and we can confirm that the dyad of distal markers (487 & 1061) is more frequently found at the tip of the protruding bars than those closer to the cXII centromere (194 & 450). Having said that, the *tetO:487* is found at the tip more often than the telomere, and the latter only goes there when the protruding bar is long enough and at later time points in the Nz incubation. In addition, and despite they are less frequently found, it is noteworthy that centromeric regions are found at the tip sometimes. This leads us to propose a model in which the driving force that forms the loop from the flare resides in the rDNA itself (with a bias towards the distal part), and that later, as the nucleus expands and the protruding rDNA bar becomes larger, the flare recoils the rest of the chromosomal arm (either the proximal or the distal part; Fig 8 in MS-R1). This model also fits well with time-course experiments we present in the revised version. Because the source of NE outward growth resides in the rDNA, we did not find worthy exploring the role of telomere attachment in these phenotypes.

3.3. We have 3D reconstructed mid-M nuclei with the horseshoe loop (Net1), the *tetR/tetOs* and the NE (Sec61 and Nup49) and found examples of bilobed nuclei with a second and smaller lobe that overlaps with the main lobe (Figs 3H, 3I and S3K; movie 6 in MS-R1). We also provide new and clearer examples where bilobed nuclei arrange in a way so that they can appear as doughnut/ring-like nuclei in 2D projections (Fig 3G in MS-R1).

4. The NE expand in the region adjacent to the nucleolus under conditions of Pah1 inactivation (e.g. *spo7Δ*, *pah1Δ* etc). Examining whether the *spo7Δ* flare, for example, is also associated with the vacuole could add novel content to this study. Either outcome (namely associated or not associated) would be interesting.

In this case, we have constructed and used a Nem1-aid strain, which renders a *nem1Δ* phenotype in auxin, but with the advantage of better controlling when Pah1 activity drops, so

cells do not have to be without its activity through many generations. We have studied this in detail and found that the nucleolus extension ("flare") in G2/M cells is not like the horseshoe loop observed in Nz (Figs 7D-G and S7G in MS-R1). In fact, it rather looks like either a "closed" class I loop or an intermediate protruding bar that is somehow unable to complete the bending to mature into a class II loop. As with non-Nz mid-M arrests, we provide data (the Nem1 depletion plus Nz experiment) that points out that the presence of microtubules is the brake for such final reshaping. As for the flare-vacuole interaction, we have found that vacuoles are also juxtaposed to these nem1 projections (Fig 7D, E).

5. Another observation that can be expanded to enhance the novelty of the study is that the addition of rapamycin causes the loop to disappear. It's implied (but not shown directly, which is an issue because rapamycin does many things) that this is due to lack of phospholipid synthesis. That continuous phospholipid synthesis is needed to maintain the loop is novel and interesting: what happens to the loop when phospholipid synthesis is blocked? Specifically—does the membrane get reabsorbed into the nucleus, leading to a large nuclear surface area? And why is there a need to continuous phospholipid synthesis once the loop has already formed?

We are sorry it was understood that we implied that the sole cause of loop contraction upon rapamycin addition was inhibition of phospholipid synthesis. We had shown in a previous paper that rapamycin, and TORC1 inhibition by other means, causes loop contraction in Nz (Matos-Perdomo & Machín, 2018a). In MS-R1, we included data on this same phenotype but adding more labels, and thus comparing the loss of the rDNA horseshoe with the shape of the nucleus and the NE. As the reviewer states, TORC1 inhibition by rapamycin alters many processes that, directly or indirectly, can cause dramatic changes in the rDNA/nucleolus, including inhibition of transcription by RNA pol I, activation of autophagy (e.g. micronucleophagy), etc. In fact, there are many previous reports that connect TORC1 and the nucleolus morphology under the microscope in cycling yeast cells (reviewed in (Matos-Perdomo & Machín, 2018b, 2019)). We have shown in this manuscript that inhibiting one of the processes under TORC1 control, specifically the phospholipid synthesis, impinges on the rDNA loop and nuclear morphologies in mid-M arrests (Figs 7H, I and S7J). We do not exclude that other processes mentioned above could act concomitantly and in a synergistic way. It would be interesting in future studies to weigh up individually these other TORC1 targets in loop maintenance.

To specifically address the question of whether the horseshoe loop maintenance needs continuous NE synthesis, we have now interfered with fatty acid synthesis once the loop has been formed (nocodazole first then Fas1-aid depletion). We have found that inhibition of fatty acid synthesis has a modest effect on the loop, and we did not observe membrane reabsorption (Fig 7I in MS-R1).

6. The authors conclude that "the absence of microtubules is a prerequisite to acquire the horseshoe rDNA loop that wraps around the vacuole". An alternative possibility is not that microtubule per se inhibit loop formation, but that in a mid-mitosis arrest in the presence of microtubules, the nucleus traverses the neck, and as a result that nucleolus no longer has access to the vacuole. Therefore, what is critically missing in this analysis is visualization of the rest of the nucleus, to determine if, under these conditions, the nucleolar region still extends away for the rest of the nucleus. If it does, but it is not associated with the vacuole, this raises questions regarding the physiological relevance of the loop-vacuole association. Specifically, what would be the physiological conditions that would require loop-vacuole interactions only where there are no microtubules?

Well, what the reviewer suggests is what we actually observed and described in Figures 4D and 4E in MS-v0. Indeed, flares still position next to vacuoles, even when the vacuole resides in the cell body opposite to the bulk mid-M nucleus. Thus, the rDNA transverses the neck (the rest of the DNA mass was stained with DAPI in those experiments). In fact, this finding strongly supports that there is a direct interaction between the rDNA flare and the vacuole (not a mere co-localization because of their relative cell occupancy), which is further supported by the hook-like morphology the rDNA sometimes adopts around the vacuole. In addition, the vacuole may provide the scaffold for this rDNA transversion, explaining this astonishing phenotype. Anyhow, with the rDNA still connected to the vacuole but with the nuclear position and morphology so different in these non-Nz mid-M arrests, we believe that microtubules prevent the horseshoe

loop just by interfering with its maturation from the flare. This was already stated in MS-v0. We now explain this better, but also provide new data that strongly support our claim:

- An improved and multilabelled *cdc16-aid* strain, where we showed that the rDNA that transverses the neck is unambiguously connected to the vacuole, and can even bend to follow the vacuole into the second body (either the mother or the bud) (new Figs 6D, E and S6D in MS-R1).

- A new strain where cells get arrested in G2/M with a monopolar spindle (*cin8-aid kip1Δ*). In this strain, there are microtubules and at least 40% of the cells appear as dumbbells with a single nucleus in the cell body (not across the neck), better mimicking the spatial distribution observed in Nz. Despite this, the horseshoe loop was not observed (new Figs 6B, 6H and S6H).

Minor issues:

1. The title does not reflect the findings in the study, and in particular the circumstances under which the NE forms the flare/loop in budding yeast. Moreover, as indicated above, the current title is an interpretation that is not based on conclusive evidence. A more appropriate title would be something along the lines of "Nuclear envelope expansion during a mitotic delay in budding yeast is associated with the vacuole". That the rDNA is in this loop was already shown as is not novel to this study.

We believe our title was in accordance with the basis and the aims of the study. As stated above, the reviewer ought to consider the rDNA-centric viewpoint of our study as well as the questions we wanted to address at the beginning. Only at a later stage, and after realizing about the non-nuclear nature of the SUL, we noted the connection with the NE expansion nicely described by the Cohen-Fix lab. Having said that, and also taking into account our new 3D data on NE reshaping, we now propose a new title which could fit better with both points of view: "The vacuole shapes the nucleus and the ribosomal DNA loop during mitotic delays."

2. First paragraph of the results: Walters et al 2014 measured the nuclear/cell volume ratio of cells with mitotic flares/loops and found it to be the same as in cells without a loop.

Again, the argument of nucleus:cytoplasm ratio in that paragraph is based on the rDNA loop centric viewpoint we began our work. As an introduction to the matter, which is later in the MS developed until realizing about the NE connection, the two citations are reviews that include many works from different labs.

3. The observation that they mitotic flare/loop is independent of the number of rDNA copies is consistent with Walters et al 2014, who showed that the loop forms even where there are no rDNA repeats.

Respectfully, we understand that the data they provided in the referred paper point out the opposite; the rDNA loop does not form in that strain despite there are flares. In any case, we do not think that that experiment is comparable to ours. In the strain they used, there are multiple copies of the rDNA unit (*RDN1*). Despite they are not aligned in a chromosomal array, and are episomal instead, these multicopy *RDN1* plasmids can have as many copies as a WT array (~150), or even more as estimated for other two-micron plasmids (not to mention it might be rather variable within the population). Even if there were fewer copies, the fact that there is no organized chromosomal array make this strain unsuitable for the comparison we undertook in Fig 1 in MS-v0 (now Fig 2I, J and S2 in MS-R1).

Also note we have now analysed the rDNA morphology in a mutant for the rDNA attachment to the NE as a reply to reviewer #2. We found that loops, SULs and protruding rDNA bars (flares) are still present in that strain (Figs 7J and S7L).

4. Sections recapitulating published data (see above, for example Figure 2A through K, Figure 4, Figure 6F and possibly other Figures/panels) should be removed.

We believe these figures should be maintained in our paper, although some of them have been updated, replaced by improved versions, or moved to supplemental information. We nevertheless acknowledge the results obtained in Cohen-Fix lab when they are equivalent, which is in most instances. Having said that, please note that (i) a full horseshoe loop with the hole (SUL) in the middle was not shown before; (ii) we have largely looked at the rDNA (i.e., the NOR) instead of other parts of the nucleolus; (iii) Net1 binds specifically to the rDNA, whereas Nsr1 does not, and not only labels the pre-rRNAs but also can bind to uncapped telomeres, G-DNA and even mRNA; thus, Net1 appears more specific for the purpose of our study; (iv) many of the referred Figs include multilabelling (up to 3-4 morphological markers), which add new layers of information to other previous phenotypes described with fewer markers; and (v) the flare they described is just one out of multiple NE reorganizations that occur upon mitotic arrests and can lead to the horseshoe loop. We present these other NE modifications in this revision and provide schematics in the last figure (Figs 3, S3, 5, S5 and 8 in MS-R1).

5. The values for non-rDNA region compaction are meaningless because the path of the DNA cannot be traced and the assumption of a straight line is unfounded.

This is already pointed out in our paper, and it is the reason we refer to the obtained values as “apparent compaction ratio” (ACR). We agree that a precise compaction ratio needs of multiple coordinates, ideally drawing the whole chromosome line (as in the rDNA). However, note that the chiefly reported reference for chromosome compaction in mitotic yeast cells is based in measurements along a few unicolor points that are also far apart (> 100 Kbps) (Guacci *et al*, 1994). Indeed, all measurements should be ideally corrected by the new knowledge we provide on the nuclear 3D structure (ours and all previous studies undertaken in Nz). Still, we think it is worth maintaining these measurements for comparisons with the previous literature, despite they are clearly unrealistic on the basis of the new findings. Anyhow, they are good enough to pinpoint that the cXII telomere is closer to the distal rDNA flank than what is expected for a straight line with a nominal compaction ratio. This fits well with the 3D model we propose for the bilobed doughnut-like nucleus.

6. Figure 3: could the different shapes of the rDNA reflect how long these cells were arrested in nocodazole? See Witkin et al 2012 for changes in rDNA shape over time as cells arrest in nocodazole.

We have now performed time-courses and movies of bar/flare and loop/doughnut formation after Nz addition (Figs 1D-F, 5A-C, 5G and 5H; movies 7, 9 and 10 in MS-R1). These data have allowed us to propose models of morphological transitions after Nz addition (Fig 8A in MS-R1).

7. The stationary phase experiment is not informative in the context it is presented: it has already been established that in cycling cells the flare/loop is specific to mitosis (here and mostly in published data), and in stationary phase the nucleus adopts a completely different form. Without delving into that just showing the nucleolus does add much.

As stated previously, the purpose of this experiment was two-fold. On the one hand, to test whether massive vacuoles could deform the nucleus so that an rDNA loop can be attained out of G2/M. On the other hand, if the vacuolar pressure was dominant over TORC1 or, alternatively, over the mid-M phase (for this, this experiment must be interpreted in conjunction with those of rapamycin). We are sorry we could not explain this properly in MS-v0; we have now done it now.

8. The discussion contains sections related to rDNA condensation which is out of place for this study, as no experiments in this regard were presented.

We would like to remind again the aim that drove our attention to carry out this study, which was purely rDNA-centric: We were interested in how the extraordinary horseshoe-like rDNA loop could be mounted. The rDNA loop has been, and still is, a matter of active research, and is considered one of the best models of chromosome condensation in *S. cerevisiae*. Recurring literature use the loop to address critical questions about the function and interplay of condensin, cohesin, Top2, etc. We do believe that showing that the vacuole is a main driver in loop formation will have profound implications in those working in chromosome structure. We

did not include experiments in this direction because we had already showed in a previous paper that the Net1 rDNA loop is not formed in condensin-aid (*smc2-aid* and *ycs4-aid*) in the presence of IAA (Matos-Perdomo & Machín, 2018a), recapitulating a large bunch of previous papers that have used condensin ts alleles and have looked at the rDNA by FISH.

9. Please number all pages- it will make it much easier to comment on specific statements, not to mention easier to read when printed.

Sorry for this forgetfulness. We have now corrected this.

Reviewer #2 (Comments to the Authors (Required)):

This paper by Matos-Perdomo et al. explores the nature of a previously described observation that ribosomal DNA (rDNA) becomes extended and forms a "horseshoe-like loop" upon treatment with nocodazole (Nz). This paper is an in depth and rigorous exploration of the biogenesis of these structures. They find that the length of the rDNA loop does not depend on the number of repeating rDNA subunits. They use superresolution microscopy of tetO arrays to demonstrate that the loop is comprised of the entire length of two sister chromatids with their telomeres flanking both ends of the loop. They then show that the rDNA loop wraps around the vacuole, which requires depolymerized microtubules, active TORC1, and phospholipid synthesis. Interestingly, the loop still wrapped around the vacuole when the only established membrane tethering complex between the nucleus and the vacuole was abolished. Finally, they show that the formation of the rDNA loop requires cells to be in the G2/M stage of the cell cycle. Overall, the quality of the data is good, and the conclusions are all appropriate and consistent with the data. I suggest only a few minor experiments, which are listed below:

1) The observation that the formation of rDNA loops requires phospholipid synthesis suggests that the aberrant synthesis of membrane adjacent to the nucleolus is responsible for the extended structure the rDNA adopts. It remains to be shown, then, whether the rDNA plays any role in the formation of the loops. I suggest disrupting the association of the rDNA to the inner nuclear membrane by deleting the CLIP complex (Heh1 and Nur1) and determining if the loop still forms and whether or not the rDNA is in the loop.

Indeed, this is an interesting question that we have now addressed with the *heh1Δ nur1Δ* double mutant. We have found that nuclear extensions, protruding rDNA bars, horseshoe loops and SULs are still present (Figs 7J and S7L in MS-R1). It is true that horseshoe loops are slightly less frequent than in a wild type strain, but the drop is relatively mild taking into account that rDNA-NE tethering is expected to be disrupted in such double mutant. This means that rDNA-NE tethering is not a prerequisite for the extraordinary phenotypes seen in Nz mid-M arrests.

2) The TEM images presented throughout the paper (in particular Figure 2J) are much too small to make out any detail and should be made larger so that the membranes can be more easily visualized.

We have now enlarged these images (Fig 3F in MS-R1) and included a third supplemental panel with the inner and outer nuclear membranes pointed by arrows, and the inner and outer faces of the double NE in the loop highlighted in different colours (Fig S3I in MS-R1).

3) "TORC1 complex" is redundant. TORC1 stands for target of rapamycin complex 1 so "complex" can be removed.

We have corrected this, thanks.

Reviewer #3 (Comments to the Authors (Required)):

Matos-Perdomo et. al., report interesting distinct morphologies of rDNA loop that adopts during the mid-M arrests. The rDNA loop results in the appearance of a space under the loop (SUL) which is devoid of any nuclear component but colocalizes with the vacuole. Authors show that the formation and maintenance of the rDNA loop and the SUL require TORC1 and membrane lipid synthesis but its independent of nuclear-vacuolar junctions. While the rDNA loops are unique, the observations shown here lack explanation in several instances. Please find below major and minor comments that authors need to address.

Major comments:

Is there any pre-existing evidence for rDNA loops to get unwound and separate in two sister strands by undergoing threading and knotting? Is this process result of break and repair by reannealing?

We do not know the nature of the “apparent” knots. It is an interesting observation that we describe here but we believe is out of the main article scope. We think that these threads and knots need to be observed with other complementary approaches as well. It is worth pointing out, though, that these short tracts of two “resolved” threads were present in just a subpopulation of loops, they were not always visible. By contrast, the spring-like nature of the rDNA in bars and loops was generally evident.

Having said this, we doubt that the observed knots represent intermediates of DNA repair processes as Nz should not cause DNA damage by itself. Instead, we favour that they may be unresolved topological structures such as catenations; at the mid-M arrest, neither condensin nor Top2 has reached its peak of activity, expected for mid-anaphase. Also, many of these crossovers result from the visual perspective with which we are looking at the loops in 2D projection. We have now included examples in the new Fig 1 in MS-R1 with the most frequent rDNA morphologies in a Nz arrest, and show that there is transposition between them simply by changing the viewing angle in 3D loop reconstructions. Crossovers and 8-like morphologies are also more frequent shortly after Nz addition, when rDNA loops are still small and packed.

Are the constrictions in the CSM images potential sites for sister chromatids to break and repair?

This relates to the previous question. In this case, we believe that the most likely explanation of these constrictions is that they connect two tracts of more packed rDNA units. Alternatively, they could be tracts that are slightly out of plane during sequential z-plane confocal acquisition. We showed these constrictions and the beads because it is clearly visible by both confocal and wide field. However, the spring/spiral-like nature of the rDNA in bars and loops is also a remarkable feature. With enough resolution in z, it may be possible that all constrictions and beads resolved into a 3D spring, but this would require a superresolution technology we do not own.

In general, Sec61 staining is not convincing, how do you differentiate external and internal faces of rDNA loops?

A problem we had in MS-v0 was that Sec61-xFP was quickly photobleached under wide field fluorescence. This was a real issue with Sec61-CFP but it also occurred with Sec61-YFP. This led us to take single z planes of Sec61 and then inferred that if we saw Sec61 labelling part of the internal face of the loop (as it was), this would need to occur all along the internal face. That is the reason there are no pictures with an entire labelling of the internal and external faces since we would have needed both an exceptionally planar horseshoe loop and a perfect front view.

What we have done now to solve this issue is to use confocal microscopy, which happens to diminish Sec61 photobleaching to a great extent, together with 3D reconstructions. We have both replaced the previous Sec61 photos in the main figures and added new examples (see new Figs 3B-D, H, I; S3F; movies 5 and 6 in MS-R1). Now, it is very clear the presence of Sec61 in the internal face.

Fig 2G, 2H, 2I, 2K are not convincing. Better representative images are required. Also, include graphs of scanning intensity on a line across the image to indicate co-localization.

We have now replaced these photos with those obtained by CSM. We have also measured co-localization as requested (Fig 3B-E; S3F and 4A in MS-R1).

Fig 2J, 3G- highlight the double membrane.

As also requested by reviewer #2, we have made a larger version of these TEM images and included a third supplemental panel with the inner and outer nuclear membranes pointed by arrows, and the inner and outer faces of the double NE in the loop highlighted in different colours (Fig S3I in MS-R1).

Please confirm nuclear flares in Fig 3. using nuclear protein PUS1.

We know that xFP-Pus1 has been used in many studies as a bona fide reporter of the nucleoplasm, despite having a biological role in chemically modifying tRNAs. We believe that the nucleoplasm markers we have used are more neutral; neither tetR-NLS-YFP nor NLS-GFP are *S. cerevisiae* proteins. We have now included new examples of nucleoplasmic labelling with these two markers in confocal microscopy. Note how the nucleoplasm follows the shape drawn by NE markers.

Can you test the role for dismantling of spindle apparatus in rDNA horseshoe loop formation without microtubule depolymerization, may be spindle keeps the NE from extending and bending?

Yes, we have now provided new experiments in this regard. What we have done is to interfere with the formation of a bipolar spindle by conditionally degrading the key kinesin 5 Cin8 in a *cin8-aid kip1Δ* double mutant. We have found that the horseshoe loop does not form in monopolar spindles when the bulk of the nucleus remains in one cell body (note that the mid-M arrest in this double mutant is a mixture Nz and *cdc16*) (new Figs 6B, 6H and S6H in MS-R1). This complements previous (and now improved data) with the *cdc16-aid* bipolar spindle (Figs 6B-E and S6D in MS-R1), and strongly suggest that microtubules per se, not a proper bipolar spindle, hinder loop formation. We discuss about this as it may point towards an uncharacterized role of microtubule cytoskeleton in the maintenance of a proper nuclear shape in the context of NE expansion.

Interestingly, the rDNA length was slightly larger in the straight line than in the horseshoe loop, pointing out that the rDNA is even more stretched in such configuration (Figure 4G). However, in galactose, a condition needed to active the SAC in the PGAL1- MAD2-MAD3 strain, the line was shorter (Figure 4H) --- little too much of interpretation. what is n used for counting this?

The strengthened rDNA in *cdc16* mid-arrest (in comparison to Nz) is likely a result of having the interacting vacuole and the bulk of the DNA mass in different cell bodies, with the rDNA going through the neck. We now provide new data with an improved *cdc16-aid* strain that support this vision (Figs 6C-E and S6D in MS-R1). On the other hand, the length of the rDNA loop likely depends on how active TORC1 is. Galactose, used for Mad2-Mad3 overexpression, is a carbon source which makes cell grow more slowly than glucose. Now we have compared the reference Nz arrest in glucose versus in galactose. We have found that galactose gives rise to shorter horseshoe loops (Fig S6F and G in MS-R1).

Nz+rapamycin: Can you separate the effects of the absence of MT from inhibition of TORC? Add rapamycin first and then nz; so even though TORC is absent, mt is disassembled. Is the concentration the same for nz and rapamycin concurrent treatment vs step by step?

The drawback of such experiment is that rapamycin arrests cells in G0/G1 (Lorenz & Heitman, 1995; Barbet *et al*, 1996; Zaragoza *et al*, 1998); and our own data (<5% of cell dumbbells if

rapamycin is added just 1h before Nz). As we showed in MS-R1 (Figs S7A-C, S9 & S10), the mid-M arrest is a requirement for the rDNA loop.

With Regards to the concurrent (Nz+rap) or step by step (Nz->rap) treatments, concentrations were the same. We have also shown before the effect of adding rapamycin after Nz treatment (Matos-Perdomo & Machín, 2018a).

What happens to rDNA loops in mid-M arrested *nem1* mutants?

This question somewhat relates to reviewer #1's point 4. For this particular case, we have used a *Nem1*-aid strain, which renders a *nem1Δ* phenotype in auxin, but with the advantage of better controlling when *Pah1* activity drops, so cells do not have to be without its activity through many (>40) generations, which may cause a starting point with already deformed nuclei. We have done the requested experiment and found that depletion of *Nem1* leads to nuclear extensions that harbour the rDNA in growing cells (Figs 7D-G and S7G in MS-R1). However, the rDNA is rarely a horseshoe; most of the times is a protruding bar or a closed loop. Once Nz is added to these cultures, horseshoe loops appear. This shows that *nem1* and Nz have additive effects (Fig 7F and G).

Figure 5I- For partial FAS-aid degradation experiment authors show that rDNA looping is not affected but only NE extension is affected. Please explain this result. Isn't there a NE around the rDNA handles visible in the representative image?

We are not saying that rDNA horseshoe loops are not affected in the same way as NE extension; in fact, rDNA horseshoes are absent under fatty acid synthesis inhibition. We believe that this misunderstanding comes from the fact that small loops (<2 microns) are present in cycling cells (see (Matos-Perdomo & Machín, 2018a); figure S1). We now explain this better from the beginning (new Fig 1 in MS-R1), emphasizing the difference between horseshoe loops and other small, packed loops. To sum up, the small loops/handle we showed after *Fas1* degradation are not horseshoe loops of a mid-M arrest, their size is quite small and are not surrounded by NE on both sides (i.e., there is no SUL). We have now pointed these tiny loops in the corresponding figure and described this in the main text and figure legend.

Fig 6E- first two bars are explained in the results, what is the third bar showing? It seems like that condition is rescuing the defect. Need explanation.

Sorry, that bar corresponds to a control strain (*net1*-aid but without the *Ostir1*), which is noted on the top. We have now explained this (in MS-R1 is the Fig S9D).

Fig 6F - the rDNA signal is still very close to the vacuolar boundary even though you don't see the extension. Please explain.

Yes, we are not saying that the nucleolus losses its position adjacent to the vacuole in this mutant; it is still there. The point is that the enlargement and/or fragmentation of the vacuole that occurs upon depletion of *Cdc28* do not cause rDNA looping by itself. Note however that, as a result of having new vacuole mutants, we have opted to remove *cdc28*-aid experiments in MS-R1. This is explained in detail right below (to your own final point).

The authors should also discuss more about the sequence of events. NE extension providing space for the rDNA knots/beads to get accommodated or the rDNA loop is a cause for forming more NE around them.

We now provide experimental evidence about these issues, partly as a response to concerns raised by the two other reviewers. We now show by time courses and videomicroscopy how nuclear extension and loop formation occurs (Figs 1D-F, 5, and S5; movies 7, 9 and 10 in MS-R1). It turns out to be more complicated than anticipated, with at least three co-existing paths that converge into the formation of the horseshoe loop. As a result, there are two major classes of loops. All is summarized in Fig 8 in MS-R1. Besides, we do not exclude other ongoing processes (e.g. rDNA transcription, condensin, etc.) playing a concomitant role in this sequence of events (please see reply to Reviewer #1 major point 5).

Is vacuolar morphology and function important for this? Authors can check by using vps mutants defective in vacuolar fusion and vma mutants defective in V-ATPase function.

This question also relates to reviewer #1's point 1. Yes, we have now made several strains with mutations for several genes involved in vacuolar physiology, morphology and/or inheritance, including *vma1Δ*³. So far, the loop was still present in most mutants but *vma1Δ* and the double mutant *vac17Δ pep12Δ* (Figs 7L and S8). The latter two suggest that proper vacuole physiology is important for the establishment/maintenance of the rDNA loop. We include these new data, but we also discuss whether the effect of these mutants is direct or indirect as vacuole mutants are also expected to affect the TORC1 pathway (Jin & Weisman, 2015; Wilms *et al*, 2017).

Minor comments:

If NVJ is not important for this interaction, it's interesting to still see the rDNA cluster very close to the vacuole. What keeps it close to the vacuole. Please explain.

We do not know and indeed it is an interesting topic to pursue in future works. It might relate to the new Sec61 ladle we observed in 3D reconstructions (Figs 3I and S3H; movies 5, 6 and 8 in MS-R1). Alternatively, it might relate to cytoskeletal (actin?) engagement between the two organelles ((n)ER-vacuole) (Hamasaki *et al*, 2005; Liu *et al*, 2022); this is something worth studying in future works as well.

Authors mention "this chapter" in the mss several times. I think it is not suitable for a research article.

We have corrected this, thanks.

I would suggest rearranging some data from main figures to supplementary figures. especially the *cdc14/cdk28* data is not that strong at this point so perhaps can be moved to supplement.

We have moved Cdc14 data to supplemental (Fig S9 in MS-R1), as requested, and we chose to remove the *cdc28-aid* data. As for the latter, there are several reasons for this decision: (i) we feel that *cdc28-aid* (G1 arrest) is somewhat out of the scope of the MS-R1; (ii) the reasons we argued for its inclusion in MS-v0 (i.e., increase in vacuolar size, fragmentation, etc.) are now better covered with the new vacuole mutants we have studied in MS-R1 (Fig 7L), which, in addition, were tested in Nz mid-M arrests; (iii) besides, the effect of the drop of CDK/Cdc28 activity in Nz-treated cells is already addressed by the ectopic release of its inhibitor Cdc14 (Fig S9); and (iv) the *cdc28-aid* was hypomorphic (growth as chains of cells) and, despite it appears it did not affect the G1 arrest after depletion, it may be better to try other non-ts non-aid alleles instead.

REFERENCES

- Barbet NC, Schneider U, Helliwell SB, Stansfield I, Tuite MF & Hall MN (1996) TOR controls translation initiation and early G1 progression in yeast. *Mol Biol Cell* 7: 25–42
- Becherer KA, Rieder SE, Emr SD & Jones EW (1996) Novel syntaxin homologue, Pep12p, required for the sorting of luminal hydrolases to the lysosome-like vacuole in yeast. *Mol Biol Cell* 7: 579–94
- Blank HM, Perez R, He C, Maitra N, Metz R, Hill J, Lin Y, Johnson CD, Bankaitis VA, Kennedy BK, *et al* (2017) Translational control of lipogenic enzymes in the cell cycle of synchronous, growing yeast cells. *EMBO J* 36: 487–502
- Bryant NJ, Piper RC, Weisman LS & Stevens TH (1998) Retrograde traffic out of the yeast vacuole to the TGN occurs via the prevacuolar/endosomal compartment. *J Cell Biol* 142: 651–63
- Burd CG, Peterson M, Cowles CR & Emr SD (1997) A novel Sec18p/NSF-dependent complex required for Golgi-to-endosome transport in yeast. *Mol Biol Cell* 8: 1089–104
- Chan Y-HM & Marshall WF (2014) Organelle size scaling of the budding yeast vacuole is tuned by membrane trafficking rates. *Biophys J* 106: 1986–96
- Cowles CR, Odorizzi G, Payne GS & Emr SD (1997) The AP-3 adaptor complex is essential for cargo-selective transport to the yeast vacuole. *Cell* 91: 109–18
- Dubots E, Cottier S, Péli-Gulli M-P, Jaquenoud M, Bontron S, Schreiber R & De Virgilio C (2014) TORC1 regulates Pah1 phosphatidate phosphatase activity via the Nem1/Spo7 protein phosphatase complex. *PLoS One* 9: e104194
- Efe JA, Botelho RJ & Emr SD (2007) Atg18 regulates organelle morphology and Fab1 kinase activity independent of its membrane recruitment by phosphatidylinositol 3,5-bisphosphate. *Mol Biol Cell* 18: 4232–44
- Fuchs J & Loidl J (2004) Behaviour of nucleolus organizing regions (NORs) and nucleoli during mitotic and meiotic divisions in budding yeast. *Chromosome Res* 12: 427–38
- Guacci V, Hogan E & Koshland D (1994) Chromosome condensation and sister chromatid pairing in budding yeast. *J Cell Biol* 125: 517–30
- Hamasaki M, Noda T, Baba M & Ohsumi Y (2005) Starvation triggers the delivery of the endoplasmic reticulum to the vacuole via autophagy in yeast. *Traffic* 6: 56–65
- Jin Y & Weisman LS (2015) The vacuole/lysosome is required for cell-cycle progression. *Elife* 4
- Lavoie BD, Hogan E & Koshland D (2002) In vivo dissection of the chromosome condensation machinery: reversibility of condensation distinguishes contributions of condensin and cohesin. *J Cell Biol* 156: 805–15
- Lavoie BD, Tuffo KM, Oh S, Koshland D & Holm C (2000) Mitotic Chromosome Condensation Requires Brn1p, the Yeast Homologue of Barren. *Mol Biol Cell* 11: 1293–1304
- Liu D, Mari M, Li X, Reggiori F, Ferro-Novick S & Novick P (2022) ER-phagy requires the assembly of actin at sites of contact between the cortical ER and endocytic pits. *Proc Natl Acad Sci U S A* 119: 1–11
- Lorenz MC & Heitman J (1995) TOR mutations confer rapamycin resistance by preventing interaction with FKBP12-rapamycin. *J Biol Chem* 270: 27531–7
- Machín F, Torres-Rosell J, Jarmuz A & Aragón L (2005) Spindle-independent condensation-mediated segregation of yeast ribosomal DNA in late anaphase. *J Cell Biol* 168: 209–19
- Matos-Perdomo E & Machín F (2018a) The ribosomal DNA metaphase loop of *Saccharomyces cerevisiae* gets condensed upon heat stress in a Cdc14-independent TORC1-dependent manner. *Cell Cycle* 17: 200–215

- Matos-Perdomo E & Machín F (2018b) TORC1, stress and the nucleolus. *Aging (Albany NY)* 10: 1–2
- Matos-Perdomo E & Machín F (2019) Nucleolar and Ribosomal DNA Structure under Stress: Yeast Lessons for Aging and Cancer. *Cells* 8: 779
- Nakashima A, Maruki Y, Imamura Y, Kondo C, Kawamata T, Kawanishi I, Takata H, Matsuura A, Lee KS, Kikkawa U, *et al* (2008) The yeast Tor signaling pathway is involved in G2/M transition via polo-kinase. *PLoS One* 3: e2223
- Wang X & Proud CG (2009) Nutrient control of TORC1, a cell-cycle regulator. *Trends Cell Biol* 19: 260–7
- Wilms T, Swinnen E, Eskes E, Dolz-Edo L, Uwineza A, Van Essche R, Rosseels J, Zabrocki P, Camerani E, Franssens V, *et al* (2017) The yeast protein kinase Sch9 adjusts V-ATPase assembly/disassembly to control pH homeostasis and longevity in response to glucose availability. *PLoS Genet* 13: e1006835
- Zaragoza D, Ghavidel A, Heitman J & Schultz MC (1998) Rapamycin induces the G0 program of transcriptional repression in yeast by interfering with the TOR signaling pathway. *Mol Cell Biol* 18: 4463–70

End notes:

¹ This arrangement would be similar to the compartmentalization observed in metazoan nucleoli, in which up to three specialized compartments can be distinguished: the fibrillary center (FC), the dense fibrillary component (DFC) and the granular component (GC). In interphase yeast, only two have been observed (FC+DFC vs GC), and by TEM; however, it appeared sensible that this could be achieved in an enlarged nucleolus during a mid-M arrest.

² It has been shown that TORC1 affects vacuolar surface area scaling independently of autophagy but through retrograde and anterograde membrane trafficking. Membrane trafficking impacts vacuole growth through fusion of vesicles with the vacuole via anterograde pathways, and contributes to vacuole shrinkage by budding of vesicles from the vacuole, known as retrograde trafficking (Chan & Marshall, 2014). For this reason, we employed two null mutants, *apl5Δ* and *atg18Δ* involved in these processes. Atg18 is involved in vacuole fission and in retrograde trafficking affecting both vacuolar volume and surface area (Efe *et al*, 2007; Bryant *et al*, 1998) while the *apl5Δ* deletion mutant blocks the alkaline phosphatase delivery pathway, which is thought to deliver vesicles directly from the Golgi to the vacuole through the anterograde trafficking (Cowles *et al*, 1997). Both *atg18Δ* and *apl5Δ* shows changes in vacuolar size (Chan & Marshall, 2014). We also checked the *pep12Δ* and *vps45Δ* mutants. *PEP12* encodes a t-SNARE, and *VPS45* encodes a Sec1/Munc18 protein. These proteins function together in the vacuole-protein-sorting pathway from endosomes to the vacuole (Becherer *et al*, 1996; Burd *et al*, 1997). Both genes have a critical role in generating a new and functional vacuole being involved in cell growth and cell-cycle progression from early G1 (Jin & Weisman, 2015). In addition, we constructed a double knockout mutant for the *PEP12* and *VAC17* genes. *VAC17* encodes a protein involved in vacuolar inheritance, acts as a vacuole-specific receptor for myosin Myo2p; Myo2p-Vac17p-Vac8p complex is involved in transport of vacuoles to newly formed daughter cell. The single *vac17Δ* mutant presents an abnormal vacuolar morphology. The *vac17Δ pep12Δ* double mutant cells show defects in the localization of Vph1-GFP and CMAC markers for the vacuolar membrane and lumen, respectively, or even to lack vacuoles in some cells (Jin & Weisman, 2015). Finally, we employed a *vma1Δ* mutant. Vma1 is subunit A of the V1 peripheral membrane domain of vacuolar H⁺-ATPase (V-ATPase). It is involved in hydrogen ion transporting and vacuolar acidification and cellular protein metabolism; it localizes to the vacuole membrane. Null mutant also has vacuolar and mitochondrial morphology defects, respiratory growth defect, reduced utilization of various carbons sources and decreased rate of vegetative growth. Moreover, it has been shown that Sch9, a downstream target of TORC1, controls the V-ATPase to maintain cellular pH homeostasis, both cytosolic pH (pHc) and vacuolar (pHv), and V-ATPase assembly (Wilms *et al*, 2017). These authors showed that protons acts as a second messenger to signal glucose availability via the V-ATPase to PKA and TORC1-Sch9 and that *sch9Δ* mutants has a partial *vma-* phenotype; in fact, *vmaΔ* cells show hypersensitivity and no growth on YPD at pH 7.5.

³ We tried: (i) a double labelling with Nvj1 and Vac8, which specifically mark NVJs. Despite the junctions were evident in G0 cells, we could barely detect them in asynchronous and mid-M cells, so we abandoned this approach; (ii) dyes for the VM (MDY-64) in conjunction with xFP labelling of the NE and the rDNA. Unfortunately, the VM dyes are incompatible with live imaging as they quickly kill the cell.

⁴ In general, live imaging experiments for the formation of the loop have been challenging. We noticed four major setbacks: (1) Clear indications of phototoxicity were present in wide field (WF) fluorescence microscopy (including the arrest of bud growth and the appearance of autofluorescence by the end of the movies). Phototoxicity was present in confocal live imaging as well, though to a lesser extent. Videomicroscopy without the 405 nm laser greatly diminished phototoxicity, but at the cost of losing the CFP signal (Net1 or Sec61). (2) Photobleaching was a serious problem as well, especially for the CFP (Net1 or Sec61). To mitigate the problems related to both phototoxicity and photobleaching, we had to reach a compromise between signal quality (exposure, laser intensity, CFP signal), number of z-planes to take, and time between frames. The number of z-planes is a critical parameter because of the 3D complexity of the morphologies we report here. For 3D reconstructions we often employed 30-50 z-planes over 5 to 8 microns in depth (0.15 μm thick); however, for movies, we had to take only 4 z-planes at the most (many times just a single z-plane). (3) Adding to the extreme complexity of 3D interpretation and the limitation of z planes in movies, the nucleus and vacuoles were extremely dynamics in Nz. We already had issues for immobilizing cells in YPD on the surface of Lab-Tek chambers with concanavalin A but, on top of that, organelles within Nz-treated cells were highly mobile. In particular, the nucleus often changed several times its spatial orientation and went out of focus during filming, leading us to disregard many putative useful cell examples. (4) rDNA loops were smaller than those observed in end point experiments and in unexposed Lab-tek fields, suggesting that imaging with LED/laser excitation, even in subphototoxic conditions, give rise to stress that affect loop formation. Contraction of the loop just by live imaging of mid-M arrested cells was already present in the movies we included in MS-v0 (movies 3 and 4), and how sensitive the loop is to environmental stress was studied by us in a previous report (Matos-Perdomo & Machín, 2018a). To sum up, we include long-term (3h) movies of loop formation, but we are aware of their

limitations, and we want to emphasize this here. This is one of the main reasons we chose to complement time-lapse live imaging with time-course experiments.

⁵ We tried to make strains with double *lacO/tetO* labelling but we were unsuccessful in constructing a stable *lacO* plasmid to target integration into the rDNA flanks.

July 20, 2022

RE: Life Science Alliance Manuscript #LSA-2021-01161-TR

Dr. Félix Machín
Hospital Universitario Nuestra Señora de Candelaria
Research Unit
Ctra del Rosario, 135
Santa Cruz de Tenerife 38010
Spain

Dear Dr. Machín,

Thank you for submitting your revised manuscript entitled "The vacuole shapes the nucleus and the ribosomal DNA loop during mitotic delays". We would be happy to publish your paper in Life Science Alliance pending final revisions necessary to meet our formatting guidelines.

-please remove the panel J (B-J should be B-I) from your figure legend for Figure 4

A. FINAL FILES:

B. MANUSCRIPT ORGANIZATION AND FORMATTING:

****It is Life Science Alliance policy that if requested, original data images must be made available to the editors. Failure to provide**

original images upon request will result in unavoidable delays in publication. Please ensure that you have access to all original data images prior to final submission.**

The license to publish form must be signed before your manuscript can be sent to production. A link to the electronic license to publish form will be sent to the corresponding author only. Please take a moment to check your funder requirements.

Sincerely,

Reviewer #2 (Comments to the Authors (Required)):

I was already enthusiastic about the quality of the data presented in this paper and felt it should be published - the authors addressed my few concerns.

Reviewer #3 (Comments to the Authors (Required)):

The authors have appropriately addressed my comments in the revision submitted. I recommend this manuscript for publication.

July 20, 2022

RE: Life Science Alliance Manuscript #LSA-2021-01161-TRR

Dr. Félix Machín
Hospital Universitario Nuestra Señora de Candelaria
Research Unit
Ctra del Rosario, 135
Santa Cruz de Tenerife 38010
Spain

Dear Dr. Machín,

Thank you for submitting your Research Article entitled "The vacuole shapes the nucleus and the ribosomal DNA loop during mitotic delays". It is a pleasure to let you know that your manuscript is now accepted for publication in Life Science Alliance. Congratulations on this interesting work.

DISTRIBUTION OF MATERIALS:

Again, congratulations on a very nice paper. I hope you found the review process to be constructive and are pleased with how the manuscript was handled editorially. We look forward to future exciting submissions from your lab.

Sincerely,
